# Recent Advances in Pyrimidine-Based Drugs

**DOI:** 10.3390/ph17010104

**Published:** 2024-01-11

**Authors:** Baskar Nammalwar, Richard A. Bunce

**Affiliations:** 1Vividion Therapeutics, 5820 Nancy Ridge Drive, San Diego, CA 92121, USA; nbaskarphd@gmail.com; 2Department of Chemistry, Oklahoma State University, Stillwater, OK 74078, USA

**Keywords:** pyrimidines, building block, antibiotic, antifungal, antiviral, anticancer, immunological treatment, neurological disorders, anti-inflammatory, chronic pain, diabetes mellitus

## Abstract

Pyrimidines have become an increasingly important core structure in many drug molecules over the past 60 years. This article surveys recent areas in which pyrimidines have had a major impact in drug discovery therapeutics, including anti-infectives, anticancer, immunology, immuno-oncology, neurological disorders, chronic pain, and diabetes mellitus. The article presents the synthesis of the medicinal agents and highlights the role of the biological target with respect to the disease model. Additionally, the biological potency, ADME properties and pharmacokinetics/pharmacodynamics (if available) are discussed. This survey attempts to demonstrate the versatility of pyrimidine-based drugs, not only for their potency and affinity but also for the improved medicinal chemistry properties of pyrimidine as a bioisostere for phenyl and other aromatic π systems. It is hoped that this article will provide insight to researchers considering the pyrimidine scaffold as a chemotype in future drug candidates in order to counteract medical conditions previously deemed untreatable.

## 1. Introduction

Pyrimidine is an important electron-rich aromatic heterocycle, and, as a building block of DNA and RNA, is a critical endogenous component of the human body [1]. Due to its synthetic accessibility and structural diversity, the pyrimidine scaffold has found widespread therapeutic applications, including antimicrobial, antimalarial, antiviral, anticancer, antileishmanial, anti-inflammatory, analgesic, anticonvulsant, antihypertensive, and antioxidant applications [2,3,4,5,6,7,8,9]. Furthermore, pyrimidines are also reported to possess potential medicinal properties important to central nervous system (CNS)-active agents, calcium channel blockers and antidepressants [10,11]. Due to its broad biological activity, pyrimidines have piqued tremendous interest among organic and medicinal chemists. In addition to its ready availability, the pyrimidine skeleton can be easily modified for structural diversity at the 2, 4, 5 and 6 positions (Figure 1).

Some of the known commercial pyrimidine-based drugs **1**–**16** are shown in Figure 2. Due to the pyrimidine ring’s ability to interact with various targets by effectively forming hydrogen bonds and by acting as bioisosteres for phenyl and other aromatic π systems, they often improve the pharmacokinetic/pharmacodynamic properties of the drug. The pyrimidine ring has unique physiochemical attributes that have led to its widespread incorporation into drug candidates with a broad spectrum of activities. The chemical space portfolio of drugs relying on this privileged scaffold has increased at a rapid rate for a wide variety of biological targets with different therapeutic requirements.

This review article attempts to comprehensively outline the synthetic strategies employed to prepare pyrimidine derivatives as well as the biological and clinical significance of these systems for various therapeutic needs. The review covers the literature of the past two years and drug candidates are organized according to the specific medical conditions they are designed to treat: bacterial, fungal and viral infections; cancer; immunological and neurological disorders; inflammation; chronic pain; and diabetes mellitus. The sheer volume of reports over this short period is a testament to the impact and potential of pyrimidine-based compounds in drug research. Throughout the manuscript, the activities of new drug prototypes are compared with numerous commercial and experimental drugs. Many of these compounds do not incorporate pyrimidine and are not shown in the text of this review. However, as readers may not be familiar with these drugs, structures for these standards are pictured in the SI along with some of the commercial names.

## 2. Pyrimidine-Based Drugs for Treatment of Infections

### 2.1. Pyrimidines as Antibacterials

Luo et al. [12] have focused on developing a lead molecule against tuberculosis (TB), a lethal infectious disease caused by *Mycobacterium tuberculosis* which is a prevalent problem in Asia and Europe [13]. Though the current treatment protocol for this infection involves a combination regimen which is very effective, resistance has started to emerge for this treatment option. Thus, a promising antitubercular compound with a novel mechanism of action must be developed against drug-resistant TB. In this work, the authors noted that Certinib (see Appendix A), an approved antitumor drug for anaplastic lymphoma kinase, expressed antitubercular properties through a phenotypic screening approach. The compound exhibited a modest minimum inhibitory concentration (MIC) of 9.0 μM/mL against the H3Ra variant. The authors in this work were able to identify the lead pharmacophore and quickly develop a structure activity relationship (SAR) for this series of compounds.

The synthetic approach to model compounds for this study is depicted in Figure 1. Nucleophilic aromatic substitution (S_N_Ar) reaction of 2,4,5-trichloropyrimidine (**17**) at C4 with commercial 2-isopropoxy-5-methyl-4-(piperdin-4-yl)aniline (**18**) in the presence of *N*,*N-*diisopropylethylamine (DIPEA) in isopropanol at 80 °C provided **19**. Compound **19** was further substituted by anilines at C2 under acidic conditions to generate compound **20**. Ammonolysis of **19** and subsequent removal of the *tert*-butoxycarbonyl (Boc)-protecting group from nitrogen with trifluoroacetic acid (TFA) in dichloromethane (DCM) afforded **21**. Finally, Suzuki–Miyaura coupling of **19** with an arylboronic acid using [1,1′-bis(diphenylphosphino)ferrocene]dichloropalladium(II) (Pd(dppf)Cl_2_) with cesium carbonate (Cs_2_CO_3_) as the base in aqueous dioxane under reflux, followed by Boc deprotection, afforded targets **22**.

A total of 58 compounds are described in this report and the publication highlights the biological significance of these drug molecules. Compounds **23** and **24** both exhibited weak activity on multidrug-resistant *Staphylococcus aureus* (MRSA), *Mycobacterium abscessus* and *Mycobacterium smegmatis* with MIC values of 4–8 μg/mL. Drug candidate **24** displayed potent activity against the H37Ra (ATCC 25177) and H37Rv (ATCC 27294) strains of TB as well as clinical drug-resistant variants with MIC values of 0.5–1.0 μg/mL. This compound also possessed acceptable toxicity in vivo, at a high oral dose of 800 mg/kg. The pharmacokinetic (PK) properties were evaluated for **24** using Sprague–Dawley rats, and the results are summarized in Table 1. Compound **24** exhibited moderate exposure with a C_max_ = 592 ± 62 mg/mL, slow elimination with a t_1/2_ = 26.2 ± 0.9 h, a low clearance (CL) value of 1.5 ± 0.3 L/h/kg following intravenous (i.v.) administration, and promising oral bioavailability (F) with a value of 40.7%.

The Mohamady group also explored the use of pyrimidines as anti-infectives against *M. tuberculosis* [14]. TB is an airborne infectious disease which primarily targets the lungs and other organs, such as spine, kidney and brain, by alternating between the active and latent phases and challenges the immune system defense mechanism [15]. Multidrug-resistant (MDR) variants of this pathogen are on the rise for Isoniazid and Rifampicin (see Appendix A) which are first line of defense drugs [16]. In this study, these researchers tried to inhibit *M. tuberculosis* by targeting the fatty acid biosynthesis pathway of the pathogen. They specifically targeted acyl carrier protein reductase which is an essential component of the mycobacterial survival pathway. Upon disruption of this pathway, the bacteria starve, resulting in cell death and eradication of TB. In this work, the concept of molecular hybridization was used to design the desired library of compounds.

Several model compounds were synthesized according to the generalized route outlined in Figure 2. The synthesis began with a one-pot multi-component reaction of benzaldehydes **25**, ethyl cyanoacetate, and thiourea with potassium bicarbonate (KHCO_3_) in ethanol to give pyrimidine derivatives **26**. Hydrazinolysis of thiol **26** with hydrazine hydrate in ethanol under reflux afforded the 2-hydrazinyl-6-oxo-4-aryl-1,6-dihydropyrimidine-5-carbonitriles **27**. These derivatives were subsequently condensed with isatin derivatives **28** in ethanol containing drops of acetic acid to produce the isatin–pyrimidine hybrids **29**.

The inhibitory activity of compound **29** was screened against three strains of TB, including a sensitive strain (ATCC 25177 = H37Ra) as well as an isoniazid (see Appendix A)-resistant strain (ATCC 35822). Some of the promising derivatives showed an MIC of <1 μg/mL on both strains, including the MDR and extremely drug-resistant (XDR) strains. Of all of the derivatives, compound **30** exhibited the maximum inhibition of MDR and XDR *M. tuberculosis* with MICs of 0.48 and 3.9 μg/mL, respectively. On the other hand, the same compound also proved the most potent against inhibin subunit alpha (InhA) with a half maximal inhibitory concentration (IC_50_) of 0.6 ± 0.94 μM. In this paper, the authors were successfully able to co-crystallize compound **30** in the ligand-active site of InhA.

Recently, the Yang research group has worked to develop a broad-spectrum antibiotic by extending the scope of Linezolid (see Appendix A) by appending a pyrimidine ring to its framework [17]. Oxazolidinone ring cores are known for their potent activity in the antibacterial space, especially toward Gram-positive pathogens [18]. Due to prevalent resistance by the pathogens, however, there needs to be continuous improvement over current medications to develop next-generation antibiotics. The authors sought antibiotic candidates that would also possess antibiofilm activity, specifically targeting urinary tract infections. The authors hypothesized that linking a pyrimidine moiety to the Linezolid structure would improve the drug’s ability to form hydrogen bonds as well as improve the permeability of the compound.

The synthesis shown in Figure 3 illustrates the preparation of a pyrimidine-linked prototype from piperazine–Linezolid precursor **31** which was synthesized by a known method [19]. Compound **31** underwent a regioselective reaction with various 2,4-dichloropyrimidine derivatives **32** in the presence of triethylamine (TEA) in ethanol to afford intermediates **33**. These individual intermediates were further reacted with various amines under mild acidic conditions in the presence of *p*-toluenesulfonic acid monohydrate (*p*-TsOH·H_2_O) in ethanol, to eventuate C4 chloride displacement from the pyrimidine ring and afford the required pyrimidine–Linezolid structures **34**.

Compounds **34** were evaluated for their antibacterial activity against seven different strains of mostly Gram-positive bacteria including *S. aureus*, *Streptococcus pneumoniae*, *Enterococcus faecalis*, *Bacillus subtilis*, *Staphylococcus xylosus*, and *Listeria monocytogens*. Several derivatives exhibited very good activity against a subset of these organisms (MIC = 0.25–1 μg/mL) with **35** exhibiting an MIC of 0.25–1 μg/mL against all of these pathogens. Candidate **35** also displayed an MIC of 1 μg/mL against the Methicillin (see Appendix A)-resistant *S. aureus* (MRSA) and Vancomycin-resistant *Enterococcus* (VRE) bacteria. Furthermore, **35** displayed a minimum biofilm inhibitory concentration (MBIC) ranging from 0.5–4 μg/mL against a series of four bacterial strains including MRSA, VRE, Linezolid-resistant *S. aureus* and Linezolid-resistant *S. pneumoniae*. These results have led the authors to conclude that compound **35** has high potential for further development as an antibacterial drug.

Kumari et al. sought to develop an antibacterial agent incorporating pyrimidines by targeting the enzyme DNA gyrase which is a bacterial topoisomerase II [20]. DNA gyrase is responsible for DNA replication, transcription, repair and decatenation in the bacteria. The gyrase is composed of two subunits, gyrase A and gyrase B, one of which controls ATPase activity while the other works by breaking and reassembling bacterial DNA. The role of DNA gyrase is to maintain the topology of the DNA present only in bacteria, which makes it an ideal target for developing antibiotics [21,22].

In this work, the Kumari group devised a route to a family of chrysin-substituted pyrimidine–piperazine hybrids and this is shown in Figure 4. Initially, 4,6-dihydroxy-2-methylpyrimidine (**36**) was assembled from acetamidine hydrochloride and diethyl malonate in the presence of sodium acetate (NaOAc) in methanol. Conversion of the hydroxyl groups in **36** to chlorides using phosphorus oxychloride (POCl_3_) gave dichloropyrimidine **37**, which was substituted by chrysin (**38**) in a S_N_Ar reaction promoted by a potassium carbonate (K_2_CO_3_) base in *N*,*N*-dimethylformamide (DMF) to give **39**. The final nucleophilic substitution reaction of piperazinyl derivatives with **39** in the presence of DIPEA in ethanol afforded targets **40**.

The synthesized compounds were evaluated against twelve different strains of bacteria and two fungi. The compounds generally exhibited modest antibacterial and antifungal activity, except for **41**, which achieved an MIC = 6.5 μg/mL against *Escherichia coli* and a respectable antifungal MIC = 250 μg/mL against *Candida albicans*.

### 2.2. Pyrimidines as Antifungals

A patent by Li et al. strived to develop a new class of safe and non-toxic anti-infective agents against a series of fungal infections, including *Candida albicans*, *Saccharomyces cerevisiae*, and *Candida parapsilosis* [23]. The patent described the preparation and evaluation of various pyrimidine derivatives from aryl sulfonamides **42** by reaction with ethyl chloroformate using K_2_CO_3_ in acetone to afford the arylsulfonylurethane **43** as shown in Figure 5. Further reaction of **43** with various substituted 2-aminopyrimidines then produced sulfonylureas **44**.

Results in the patent indicate that some of the model compounds had antifungal activity equivalent or up to 5-fold higher than the known drug Amphotericin B (see Appendix A), and that the MIC_90_ of some of these compounds, including compound **45**, exhibited 3- to 30-fold better activity than Fluconazole (see Appendix A). The drug prototypes exhibited very strong antifungal properties against many strains of *C. albicans*, *S. cerevisiae* and *C. parapsilosis*. Compound **45** exhibited potent activity toward *C. albicans* in RPMI 1640, YNB, and YPD media and had the most promising MIC_90_ = 0.05–0.3 μg/mL. It also showed encouraging activity with an MIC_90_ < 0.05–0.1 μg/mL against Fluconazole-resistant *C. albicans*, *S. cerevisiae* and *C. parapsilosis* strains. Most of the compounds tested in this series had persistent antifungal activity which did not decrease even after 72 h.

### 2.3. Pyrimidines as Antivirals

Kang et al. advanced an interesting study on pyrimidines as drug scaffolds for anti-infectives which effectively studied acquired immunodeficiency syndrome (AIDS) caused by human immunodeficiency virus (HIV) [24]. With the advent of antiretroviral therapy, the disease classification of HIV has changed from being a mortal disease to a manageable chronic disorder. In this effort, the authors were focused on developing non-nucleoside reverse transcriptase inhibitors (NNRTIs) which are common among antiviral drugs. The prescribed first-generation and second-generation NNRTIs suffered from serious drug resistance due to mutant strains that rendered these drugs ineffective. The most common mutant strains, K103N and Y181C, arose against first-generation drugs while E138K developed toward second-generation drugs [25,26]. Apart from this, newer compounds had limited solubility profiles resulting in low bioavailability. Thus, it was imperative to create new structures with better adsorption, distribution, metabolism, and excretion (ADME) properties as leads for antiviral drugs.

In this work, the final target compounds were prepared in one step from previously reported compounds **46a–c** [27] as shown in Figure 6. Suzuki–Miyaura coupling of **46a–c** with arylboronic acids in the presence of tetrakis(triphenylphosphine)palladium(0) (Pd(Ph_3_P)_4_) and K_2_CO_3_ in DMF at 100 °C gave candidates **47** for this study.

The results were summarized for 39 compounds that were prepared and evaluated. Screening showed that most of the compounds exhibited antiviral activity (half maximal effective concentration, EC_50_ < 10 nM) against the HIV-1-IIIB strain compared with Etravirine (**2**, EC_50_ = 3.5 nM). Most compounds also exhibited activity on the double mutant strain RES056 (K103N/Y181C) with an EC_50_ = 50 nM, and compound **48** displayed the most potent EC_50_ values between 3.43–11.8 nM against a panel of wild type (WT) and resistant mutants. Notably, **48** displayed the highest potency against K103N and Y188L with EC_50_ values of 4.77 nM and 15.3 nM, respectively. From an ADME and toxicological perspective, **48** demonstrated no cytochrome P450 (CYP450) inhibition (IC_50_ > 10 μM) and had a favorable PK profile as shown in Table 2. The clearance of the compound was 82.7 ± 1.97 mL/h/kg (slightly high) after i.v. administration of 2 mg/kg with a sufficient oral bioavailability (F) of 31.8% following oral administration (p.o.) of 10 mg/kg. Finally, **48** did not show any acute toxicity in Kunming mice up to the maximum concentration of 2000 mg/kg.

Liu et al. also sought to develop a new drug against the HIV-1 reverse transcriptase (RT) for the treatment of AIDS [28]. RT has an important biochemical function for viral replication of DNA/RNA-dependent DNA polymerase and ribonuclease H [29]. Currently, RT inhibitors are mainly divided into nucleoside RT inhibitors (NRTIs) and non-nucleoside RT inhibitors (NNRTIs) [27]. NNRTIs have been key components in highly active antiretroviral therapies due to their promising anti-HIV-1 properties, high specificity, and relatively low toxicity. In this work, the authors emphasize the need for novel NNRTIs with higher anti-HIV-1 activities against resistant mutant strains and improved drug properties.

The synthesis of candidate compounds for this project is shown in Figure 7. Initially, nitroaryl halides **49a-c** were reacted with Boc-piperazine in the presence of K_2_CO_3_ in DMF at 120 °C to afford **50**. These adducts were converted to the first precursor by reduction to the aniline derivatives **51** with H_2_ and Pd/C in methanol. To prepare the second precursors, 2,4-dichloropyrimidine (**52**) underwent reaction with cyanophenols **53** (**a**: 4-hydroxy-3,5-dimethylbenzonitrile or **b**: (*E*)-3-(4-hydroxy-3,5-dimethylphenyl)acrylonitrile) in the presence of K_2_CO_3_ in DMF to generate **54**. Ether **54b** was further subjected to Buchwald–Hartwig coupling with **51** in the presence of palladium acetate (Pd(OAc)_2_), 3,5-bis(diphenylphosphino)-9,9-dimethylxanthene (xantphos), and Cs_2_CO_3_ in dioxane to produce **55b**. Boc deprotection of **55b** using TFA in DCM to afford **56b** which was acylated or sulfonated at the piperazine nitrogen to deliver the final targets **57b**.

Interestingly, among the newly synthesized compounds, **58** demonstrated significantly improved antiretroviral activity compared with the known NNRTI BH-11c (see Appendix A) against all tested HIV-1 strains. Furthermore, **58** possessed subnanomolar potency (0.1–2.6 nM) against WT and five mutant HIV-1 strains, including L100I, K103N, Y181C, E138K and F227L/V106A. Further molecular dynamics simulation studies were conducted to explain the differences between the inhibitory activity of **58** and Etravirine (**2**) against RT variants. Candidate **58** displayed improved water solubility (13.46 μg/mL at pH 7.0) compared with **2** (<1 μg/mL at pH 7.0), with an appropriate ligand efficiency (LE) value of 0.32. Moreover, **58** expressed significantly lower inhibitory activity than **2** and Rilpivirine (**3**) against CYP2C9, indicating that **58** was less likely to cause drug–drug interactions.

Compound **58** was evaluated for its PK profile in a Sprague–Dawley rat model as shown in Table 3 and demonstrated an acceptable t_1/2_, moderate clearance and favorable distribution volume after an i.v. dose of 2.0 mg/kg. When administered p.o. at a dose of 10.0 mg/kg, **58** had a poor C_max_ and concentration–time curve (AUC). Nevertheless, the oral bioavailability (F) of **58** was determined to be 1.34%. Consequently, pending further optimization, the authors tagged **58** as a promising lead compound worthy of additional study.

Chen et al. have endeavored to optimize a pyrimidine-based drug scaffold to target the influenza virus [30]. Influenza is an infectious disease of the respiratory tract and results in >300,000 deaths/year worldwide, posing a huge social and economic burden to society [31]. One way to lessen this burden is to effectively use the influenza vaccine; however, due to antigenic drift and mismatch between the vaccine and circulating strains, the vaccine is not always effective. Although neuraminidase inhibitors (Oseltamivir, see Appendix A) and RNA polymerase (RNAP) inhibitors (Baloxavir, see Appendix A) are used as first-line-of-defense drugs, resistant mutations have evolved, leading to efforts to develop new direct-acting antivirals to combat drug-resistant mutations and to identify new drugs with novel mechanisms of action [32]. These researchers reported a medicinal chemistry strategy for anchoring aza-β^3^- or β^2,3^-amino acids on a 7-azaindole ring to the RNAP subunit PB2 which has proven to be an ideal target for antiviral drug development. Benefiting from facile structural elaboration, aza-β-amino acid motifs with diverse size, shape, steric hindrance, and configuration were linked to a pyrimidine and evaluated for their antiviral activities.

The preparation of experimental compounds is presented in Figure 8 and started with 2,4-dichloro-5-fluoropyridine (**59**). After substitution of **59** at C4 by ethyl *N*-amino-*N*-(alkyl)glycinates **60** using DIPEA in THF, intermediates **61** were subjected to Suzuki–Miyaura coupling with azaindoleboronic acids **62** using (Ph_3_P)_4_Pd and K_2_CO_3_ in aqueous acetonitrile (ACN) to afford **63**. This structure was hydrolyzed using lithium hydroxide (LiOH) in aqueous THF to afford the desired acid targets **64**.

Tests on these compounds revealed that **65** (HAA-09) targets the influenza PB2_cap binding domain with potent anti-influenza virus efficacies using both in vitro and in vivo models. This drug candidate possessed high inhibition against influenza A virus polymerase and was active in submicromolar concentrations, with an IC_50_ = 0.06 μM and an EC_50_ = 0.03 μM. Compound **65** also exhibited superior antiviral activity against the Oseltamivir-sensitive A/WSN/33 and Oseltamivir-resistant H275Y variants. It showed no inhibition of the human ether-á-go-go related gene (hERG) channel, demonstrating a low risk for hERG-related cardiac repolarization (manual patch, IC_50_ *>* 10 μM) and high plasma stability (t_1/2_ > 12 h). Moreover, a subacute toxicity study was carried out in healthy mice to assess the safety profiles in vivo. Lead compound **65** demonstrated a favorable safety profile with oral administration in healthy mice at a high dose of 40 mg/kg once daily for three days. The PD read-out on oral administration for **65** indicated more than a 2-log viral load reduction and survival benefit in a mouse lethal infection model. The rapid reduction in the amount of influenza A virus in the lungs of infected mice confirmed that **65** had a direct effect on viral replication. Based on these findings, structure **65** was considered a potential PB2 inhibitor suitable for further anti-influenza drug development.

## 3. Pyrimidine Based Drugs for the Treatment of Cancer

Pyrimidines have myriad biological activities, including as anticancer pharmacophores. Zhang et al. have explored the role of pyrimidine rings as anticancer agents in breast cancer cell lines [33]. Triple-negative breast cancer (TNBC) is a heterogenous aggressive breast cancer which leads to high mortality rates due to distant metastasis and lack of efficient targeted therapeutics [34].

Focal adhesion kinase (FAK) is a non-receptor tyrosine kinase which plays a significant role in integrin-activated signal transduction by initiating a cascade of biological functions [35]. These FAK-mediated signaling pathways lead to tumor progression and metastasis by regulating proliferation through invasion and cell survival strategies. Breast cancer cells overexpress FAK kinases, which in turn activate FAK signaling pathways for cell proliferation and metastasis. Thus, inhibition of the FAK kinase could potentially slow the signaling that leads to the spread of TNBC.

The synthesis of potential FAK inhibitors is shown in Figure 9. Initial S_N_Ar reaction of 2,4,5-trichloropyrimidine (**17**) with 2-amino-*N*-methylbenzamide (**66**) using sodium bicarbonate (NaHCO_3_) in ethanol afforded the monosubstituted pyrimidine **67**. A second S_N_Ar reaction on **67** by methyl 2-(4-aminophenyl)acetate (**68**) promoted by 12 N hydrochloric acid (HCl) in isopropanol yielded the 2,4-diamino-5-chloropyrimidine **69**. This compound was then condensed with 1,2,5-oxadiazole-2-oxide derivatives **70** [36] in the presence of *N*-(3-dimethylaminopropyl)-*N’*-ethylcarbodiimide hydrochloride (EDAC) and 4-(dimethylamino)pyridine (DMAP) to provide targets **71**.

The biological results in the paper pinpointed compound **72**, which exhibited FAK inhibition (IC_50_ = 27.4 nM) and displayed a strong inhibitory effect on cell proliferation with an IC_50_ = 0.126 μM. The compound further exhibited potent inhibitory effects on an MDA-MB-231 TNBC cell line but displayed a 19-fold lesser effect on non-cancer MCF10A, giving a nearly 20-fold window for cell differentiation. Importantly, treatment with **72** inhibited lung metastasis of TNBC more potently than known compound TAE226 (see Appendix A) in mice. The compound also exhibited some off-target activity by showing significant inhibition activity against matrix metalloproteinase-2 (MMP-2) and MMP-9. A pharmacodynamic effect was also observed in a BALB/c nude mouse model by inoculation with MDA-MB-231 TNBC cells in the tail vein. Once the metastatic nodules were formed, the mice were randomly injected with **72** over a period of 30 days. It was found that **72** at 15 mg/kg significantly reduced the lung tumor nodules relative to the vehicle-treated control.

Concurrent with the work of Zhang, the Badawi group was also intent on developing a drug scaffold to challenge TNBC using *N*-pyrimidin-4-ylhydrazones [37]. Though the authors mentioned breast cancer as their focus, they did not clearly define their goal in this work. The report indicates they were leaning toward the epidermal growth factor receptor (EGFR) or the estrogen receptor as possible targets for inhibition.

The preparation of prospective drug compounds for this study are shown in Figure 10. This involved reaction of cyano ester **73** and methyl carbamimidothioate (**74**) using NaOAc in DMF to prepare dihydropyrimidinone **75** by a known method [38]. Compound **75** was then treated with a series of cyclic aliphatic amines to provide **76**. Exposure of **76** to POCl_3_ and *N*,*N-*dimethylaniline at 60 °C gave derivatives **77** which were reacted with hydrazine to deliver hydrazinyl derivatives **78**. The hydrazinyl function of these structures was finally condensed with benzaldehyde (**79**) or acetophenones **81** to generate the requisite pyrimidine–hydrazone conjugates **80** and **82**, respectively.

Preliminary screening for antiproliferative activity revealed that some screened candidates exhibited nearly equal IC_50_ values of 0.87–12.91 μM in MCF-7 and 1.75–9.46 μM in MDA-MB-231 cells, and better growth inhibition activities than those of the positive control 5-Fluorouracil (5-FU, see Appendix A)) which showed IC_50_ values of 17.02 μM and 11.73 μM, respectively. Compound **83** offered the best selectivity index with respect to both MCF-7 and MDA-MB-231 cancer cells in comparison with 5-FU and elicited the highest increase in caspase 9 levels in MCF-7 treated samples, attaining 27.13 ± 0.54 ng/mL compared with 19.011 ± 0.40 ng/mL observed from a Staurosporine (see Appendix A) standard.

Research by El Hamd et al. sought to develop imidazole–pyrimidine–sulfonamide hybrids as inhibitors for the EGFR in mutant cancer cells [39]. Currently, there are many research groups across the world interested in developing inhibitors for EGFR, which plays a crucial role in many human cancers. EGFR, a member of the ErbB subfamily of tyrosine kinases, is overexpressed in many cancers, including those of the breast, colon, ovaries and prostrate [40]. Due to its impact on cancer progression, many therapies are currently approved for this target, including notables such as Cetuxiab, Pantitumumab and Necitumumab in antibody treatment (see Appendix A) and Neratinib, Gefitinib, Lapatinib, Afatinib and Vandetinib in small molecule treatment (see Appendix A).

Pyrimidine pharmacophores are well established as anti-EGFR lung cancer agents, and the authors in this work proposed to couple the pyrimidine ring with a sulfonamide core to bring dual activity against EGFR/human epidermal growth factor receptor 2 (HER2) breast cancer cell lines [41]. Additionally, the authors were also interested in developing inhibitors against drug-resistant mutant EGFR-L858R/T790M/C797S cell lines.

Figure 11 outlines a multi-component synthesis route to access *N*-(pyrimidin-2-yl) 4-(2-aryl-4,5-diphenyl-1*H*-imizazol-1-yl)benzenesulfonamides **87**. The reactions were carried out using 4-amino-*N*-(pyrimidin-2-yl)benzenesulfonamide (**84**), aryl aldehydes **85**, and benzil (**86**) with ammonium acetate (NH_4_OAc) and dimethylamine using diethyl ammonium hydrogen sulfate (ionic liquid) under reflux to afford the final targets **87**.

The compounds synthesized were screened against a panel of 60 cancer cell lines at a single dose of 10 μM at the National Cancer Institute. The results revealed 9 compounds that showed excellent cytotoxicity against all tested cell lines with growth inhibitions up to 95%. Two compounds, **88** and **89**, demonstrated inhibition against HER2 (IC_50_ = 81 ± 40 ng/mL and 208 ± 110 ng/mL, respectively), against the EGFR-L858R mutant (IC_50_ = 59 ± 30 ng/mL and 112 ± 60 ng/mL, respectively)**,** and against the EGFR-T790M mutant (IC_50_ = 49 ± 20 ng/mL and 152 ± 70 ng/mL, respectively). Both compounds induced MCF-7 cell death with a Bax/Bcl-2 expression ratio pointing to a mitochondrial apoptosis pathway. The authors are currently optimizing the active candidates to identify the most promising inhibitors for development.

The work of Zhang et al. expanded the scope of EGFR inhibitors, especially for targeting non-small cell lung cancer (NSCLC) cell lines [42]. As mutation or overexpression of EGFR is the main cause of NSCLC, it is considered the main target for treating this disease [43]. The authors focused on a fourth-generation reversible EGFR-tyrosine kinase inhibitor (TKI) by targeting the cysteine in the active binding site and focusing on the mutant EGFR cancer cell lines. Gefitinib and Erlotinib are first-generation EFGR-TKIs and have been shown to be very effective in NSCLC patients [44,45]. The gatekeeper mutation T790M in the ATP binding domain in EGFR is the primary mechanism of resistance which first develops in patients after 6–12 months of treatment. Considering this new mutant, second- and third-generation EGFR-TKIs were developed which display potent activity against EGFR-T790M while sparing WT cells.

A generalized route to the required compounds for this study is outlined in Figure 12. S_N_Ar reaction of 5-bromo-2,4-dichloropyrimidine (**90**) with various anilines in the presence of DIPEA in isopropanol afforded intermediates **91**. The pyrimidine ring of **91** was subjected to a second nucleophilic substitution with various amines (mostly aniline derivatives) in the presence of *p*-TsOH·H_2_O in butanol to afford bromopyrimidines **92**. Finally, these bromides were coupled under Suzuki–Miyaura conditions with various arylboronate esters **93** using Pd(dppf)Cl_2_ and potassium acetate (KOAc) in dioxane to furnish drug candidates **94**.

The authors screened the newly designed and synthesized 2-(phenylamino)pyrimidine derivatives from this sequence for activity against EGFR triple mutant cell lines. One compound, **95**, showed a promising IC_50_ value of 0.2 ± 0.01 μM against proliferation of the EFGR-Dell9/T790M/C797S and EGFR-L858R/T790M/C797S cell lines. The same compound exhibited a slightly higher antiproliferative activity than the commercial drug Brigatinib (see Appendix A). Most of the compounds exhibited weak activities on EGFR-WT, which indicates that the compound was selective for mutant EGFR. Compound **95** also significantly inhibited EGFR phosphorylation, induced apoptosis in EGFR-Dell9/T790M/C797S, and arrested the cell cycle at the G2/M phase. The results indicate that **95** was a potent fourth-generation reversible EGFR-TKI which warranted further study.

As with most medicinal agents, drug resistance has become an issue for Osimetinib, and this has been a driving force behind the development of EFGR inhibitors. This drug, which is currently used for NSCLC, showed drug resistance after the median survival time of 9.6 months [46]. Thus, Xu et al. determined to solve this problem by developing fourth-generation inhibitors with additional interactions between the compound and the protein to compensate for the loss of the conventional covalent cysteine interaction [47].

The 2,4-di(arylamino)pyrimidine core is a key ring scaffold for maintenance of activity in these known inhibitors of mutant EGFR kinases [48]. All compounds synthesized in this work were designed following a molecular modelling analysis of the crystal structure of EGFR-L858R/T790M/C797S (PDB code: 6LUD) using Autodock 4.2 software. The synthesis of prototype molecules is shown in Figure 13. S_N_Ar reaction between 2,4,5-trichloropyrimidine (**17**) and phenylenediamine (**96**) in ACN at −10 °C provided the C4-substituted pyrimidine derivative **97**. This compound subsequently underwent amide formation with acryloyl chloride and DIPEA in dioxane to afford **98**. Some of the pyrimidinamides were also prepared from acetic anhydride (Ac_2_O) in the presence of TEA in ethyl acetate. Pyrimidinamides **98** were subsequently coupled with various substituted anilines using Pd(OAc)_2_, xantphos, and Cs_2_CO_3_ in dioxane to furnish targets **99**.

All derivatives were evaluated for their effect on the enzymatic activity of EGFR-WT and mutant EGFR-L858R/T790M/C797S and EGFR-L858R/T790M kinases using the ADP-Glo Kinase Kit. Osimertinib was employed as a positive control. One of the inhibitors, **100**, was identified as the most favorable compound and strongly inhibited EGFR-L858R/T790M/C797S and EGFR-L858R/T790M activity with IC_50_ values of 5.51 nM and 33.35 nM, respectively. In addition, **100** exhibited stronger antiproliferative activity against NSCLC cells (H1975), expressing high levels of EGFR-L858R/T790M and Ba/F3-EGFR-L858R/T790M/C797S cells with IC_50_ values of 0.442 μM and 0.433 μM, respectively. Proliferation was inhibited by arresting the H1975 cells at the G2/M phase, promoting apoptosis of the cells, and reducing phosphorylation of EGFR and extracellular signal-related kinase 1/2 in a dose-dependent manner. The wound-healing assay data showed that H1975 migration and invasion abilities were effectively inhibited by **100** in a concentration-dependent manner. Compound **100** also expressed a 27-fold lower toxicity against normal liver cells, indicating an improved dosage safety margin. The results further suggest that this compound could be used as a competitive ATP inhibitor, as well as an allosteric inhibitor of EGFR-L858R/T790M/C797S.

A patent developed by Lee et al. featured a lung cancer subtype which is an EGFR mutation positive for NSCLC [49]. More than 50% of NSCLC patients have EGFR activating mutations. Currently, third-generation EGFR-TKIs are being explored to overcome this resistance. Osimertinib is a powerful inhibitor that suppresses EGFR mutations and T790M resistant mutations, but it causes ineffective binding and subsequent C797S resistance in NSCLC patients. When Osimertinib was administered as a front-line therapy, the most common resistance mechanisms proved to be the C797S mutation (7%) and mesenchymal epithelial transition amplification (15%) [50]. The next-generation EGFR compounds would need to inhibit Dell9/T790M/C797S, L858R/T790M/C797S, Dell9/C797S, and L858R/C797S, and be highly selective versus EGFR-WT to avoid adverse effects. The work in this patent focused on an unmet need, to develop a next-generation TKI targeting both C797S triple and double mutants. It was imperative to create a selective, next-generation inhibitor for NSCLC patients with advanced or metastatic diseases carrying Dell9/T790M/C797S, L858R/T790M/C797S, Dell9/C797S, or L858R/C797S mutations following second-line or upfront use of third-generation EGFR-TKIs.

The synthetic approach to these next-generation anticancer agents is depicted in Figure 14. Initially, amination of the commercial pyridine derivative **101** by S_N_Ar displacement of the C4 chloride using DIPEA in DMF at 90 °C provided **102**. Subsequent exposure to Buchwald–Hartwig coupling conditions with pyrimidine derivative **103** using tris(dibenzylideneacetone)dipalladium(0) (Pd_2_(dba)_3_), xantphos, and Cs_2_CO_3_ in dioxane provided products **104**.

The enzymatic biochemical assays for the EGFR kinases were reported in the patent. The assays were conducted and reported for the EGFR-WT, double mutants Dell9/C797S and L858R/C797S, and triple mutants Dell9/T790M/C797S and L858R/T790M/C797S. There were many promising compounds that had IC_50_ values in the 0.1–100 nM range but no further biological data were reported.

A patent by the Dai group disclosed the use of pyrimidines with deuterated substituents to target cyclin-dependent kinases (CDKs) [51]. CDKs are part of a subfamily of serine/threonine protein kinases which play a significant role in regulating cell cycle progression [52]. They are essential cell cycle drivers, especially CDK2, which helps cells to transition from late G1 into S and G2 phases. CDK2 plays a prominent role in proliferative pathways, which are not important for normal cell proliferation but are essential for cancer cells [52]. Selective CDK2 inhibitors might target tumors which are highly cyclin E1 and E2 expressive. Cyclin E1 is always overexpressed in human cancer. Cyclin E1 amplified ovarian cancer cell lines are sensitive to reagents that inhibit CDK2 activity or decrease cellular CDK2 protein. Some of the pyrimidine-based drug candidates in this patent specifically targeted CDK2 and offered selectivity over other kinases in treating patients with tumors.

A strategy by which to synthesize potential CDK2 inhibitors is shown in Figure 15. Compound **105** underwent a S_N_Ar reaction with deuterated 4-aminobenzenesulfonamide **106** in the presence of zinc chloride and TEA in DCM–*t*-butanol to provide **107**. Intermediate **107** was further reacted with lithium hexamethyldisilazide (LiHMDS)-derived alkoxide from tetrahydro-2*H*-pyran-4-ol **108** in tetrahydrofuran (THF) to afford **109**. Finally, removal of the *p*-methoxybenzyl (PMB) protecting group was carried out in the presence of 2,3-dichloro-5,6-dicyano-*p*-benzoquinone (DDQ) in DCM and H_2_O to furnish derivatives **110**. Some derivatives without the deuterium are also reported in this patent using the same sequence.

The most promising compounds, **111**–**113**, had IC_50_ = 1–10 nM for CDK2/cyclin E1 activity whereas for CDK1/cyclin B1 activity was nearly 10–20 times weaker offering only 20-fold selectivity. With respect to other isoforms of cyclin—CDK4, CDK6, CDK7, and CDK9—**111**–**113** offered 100–1000-fold greater selectivity. The patent asserts that these compounds were evaluated against breast, ovarian, bladder, uterine, prostate, lung (including NSCLC, SCLC, squamous cell carcinoma or adenocarcinoma), esophageal, head and neck, colorectal, kidney (including renal cell carcinoma), liver (including hepatocellular carcinoma), pancreatic, stomach, and thyroid cancers. The patent further claims that these derivatives were tested for estrogen receptor-positive/hormone receptor-positive, HER2-negative, HER2-positive, triple negative, and inflammatory breast cancer, but few results are reported from these experiments.

As in the previous entry, Zhou et al. were involved in developing a drug scaffold targeting the CDKs [53]. CDKs are important in many crucial processes, such as cell cycle and transcription, as well as communication, metabolism, and apoptosis. Deregulation of any stage of the cell cycle or transcription leads to apoptosis but, if uncorrected, can result in a series of diseases, such as cancer, neurodegenerative diseases (Alzheimer’s or Parkinson’s diseases), and stroke [50]. CDK4/6 is considered a potential anticancer drug target. To date, three CDK4/6 inhibitors have been approved; however, there is still a gap between the clinical requirements and the approved drugs [54]. Thus, selective and oral CDK4/6 inhibitors are urgently needed, particularly for monotherapy. This study investigated the interaction between Abemaciclib (see Appendix A) and human CDK6 using molecular dynamics simulations. Based on these modelling studies, a candidate compound was designed that was predicted to show a significant inhibitory effect on a human breast cancer cell line.

The strategy to prepare the designed model compound is outlined in Figure 16. Initially, Suzuki–Miyaura coupling of 2,4-dichloro-5-fluoropyridine (**59**) with pyrazolo-pyridineboronate ester **114** in the presence of Pd(dppf)Cl_2_ and K_2_CO_3_ in aqueous dioxane afforded intermediate **115**. This was followed by Buchwald–Hartwig coupling with a 4-(4-isopropylpiperazin-1-yl)aniline (**116**) under standard conditions, to furnish **117**.

The inhibitory activity of **117** (C2213-A) was validated against CDK6 using a kinase profiling radiometric protein kinase assay. The IC_50_ value for **117** was 290 nM, comparable to the estimate of 238 nM for Abemaciclib targeting human CDK6/cyclin D3. The antiproliferative activity of **117** was significantly higher than Abemaciclib (positive control) with an IC_50_ = 2.95 ± 0.15 μM. The inhibitory activity of **117** was tested against MCF-7 cells as well as other breast cancer cell lines such as T-47D, MDA-MB-452 and MDA-MB-468 and showed a better inhibitory effect than the control. The CDK4/6 inhibition by **117** and the phosphorylation of retinoblastoma tumor suppressor were assessed by a Western blot assay on MDA-MB-231 cells. Compound **117** was found to block the CDK4/6/Rb/E2F signaling pathway in a dose-dependent manner after 24 h of incubation.

Zhang et al. have identified a new anticancer drug incorporating the pyrimidine scaffold that targets microtubules [55]. Microtubules are essential structural components of the cytoskeleton and are composed of α- and β-tubulin heterodimers [56]. Due to the polymerization dynamics of tubulin, microtubules are important targets for anticancer drugs known as microtubule-targeting agents (MTAs). A total of 7 binding sites on tubulin have been found including Paclitaxel (see Appendix A), Laulimalide, Colchicine (see Appendix A), Vinblastine, Maytansine, and Pironetin, as well as a 7th binding site. To date, no tubulin inhibitors targeting the colchicine binding site (CBS) have been specifically approved for clinical application. On the other hand, one CBS inhibitor, namely ABI (**118**), has been reported to manifest nanomolar potency against multidrug resistant strains with significant in vivo antitumor efficacy [57]. Despite the excellent biological activities, ABI analogs contain a ketone group between the imidazole and the C-ring which is a metabolic soft spot susceptible to reduction by liver microsomes. Osimertinib (**1**), on the other hand, is an approved pyrimidine-containing anticancer drug for NSCLC [58]. In this work, the authors designed the series of Osimertinib–ABI hybrids shown in Figure 3 for use as MTAs to treat cancer.

The general synthetic approach to hybrids **119**–**122** is shown in Figure 17. In Figure 17-I, construction of **119** involved Suzuki–Miyaura coupling between C4 of 2,4-dichloropyrimidine (**52**) and a wide range of commercially available arylboronic acids to generate intermediates **123**. S_N_Ar reaction of **123** with 3,4,5-trimethoxyaniline (**124**) provided drug candidates **119**. In Figure 17-II, independent Suzuki–Miyaura couplings were carried out between 3,4,5-trimethoxyphenylboronic acid (**125**) and compounds **52** and 4,6-dichloropyrimidine (**127**) to furnish intermediates **126** and **128**. Each of these was further coupled with arylboronic acids to give 4-(3,4,5-trimethoxyphenyl)-2-arylpyrimidines **120** and 4-(3,4,5-trimethoxyphenyl)-6-arylpyrimidines **121**, respectively. In Figure 17-III, aldol condensation of 1-(3,4,5-trimethoxyphenyl)ethan-1-one (**129**) and a series of benzaldehyde derivatives yielded 1,3-diaryl-2-propen-1-ones (chalcones) **130**, which were reacted with guanidine hydrochloride under basic conditions to produce a library of 2-amino-3,5-diarylpyrimidine derivatives **122**.

A total of 43 pyrimidine analogs were synthesized and evaluated for their antiproliferative activity. Among these, prototype **131**, bearing a fused 1,4-benzodioxane moiety, exhibited the best potency, inhibiting four cancer cell lines including A549–lung (IC_50_ = 0.80 ± 0.09 μM), HepG2–liver (IC_50_ = 0.11 ± 0.02 μM), U937–lymphoma (IC_50_ = 0.07 ± 0.01 μM), and Y79–retinoblastoma (IC_50_ = 0.10 ± 0.02 μM). Furthermore, **131** suppressed tubulin polymerization and disrupted the microtubule network of HepG2 cells. Molecular dynamics simulations suggested that **131** blocked the cell cycle at the G2/M phase and eventually induced HepG2 cell apoptosis by regulation of G2/M related protein expression of cyclin B1 and P21. Both scratch and transwell assays have indicated that this derivative inhibited migration and invasion of HepG2 cells in a dose-dependent manner. Overall, these results indicate that **131** has potential as a tubulin polymerization inhibitor targeting the CBS and merited further investigation.

MTAs are an important chemotherapeutic class of drugs that interfere with tubulin dynamics by disrupting the formation of the mitotic spindle, arresting cell cycles, and finally promoting apoptosis of tumor cells. An investigation by Wang et al. [59] has demonstrated that, while microtubule disruption can interfere with cancer development, it can also affect normal cells, leading to two major toxicities: neutropenia and peripheral neuropathy in postmitotic neurons [60]. Although achievements have been made in clinical treatment, toxicities still limit the utility of MTAs [61]. The CBSs, located at the interface between α- and β-tubulin heterodimers [62], effectively impact protein trafficking and could serve as a key entry point for anticancer agents. Evidence suggests that CBS inhibitors can overcome drug resistance mediated by P-glycoprotein (P-gp), multidrug resistance protein 1 (MRP1) and MRP2, and destroy the vascular networks that exist in tumor tissues serving as vascular damaging agents (VDAs). For these reasons, the CBS is an attractive target for the development of chemotherapeutic drugs, including those featured in their paper. The authors have identified a novel MTA skeleton which inhibits tubulin polymerization at 5 μM. The team further optimized this series of compounds by evaluating a structure–activity relationship (SAR) derived from an X-ray co-crystal with the target.

A concise synthesis of the required pyrimidine analogs is presented in Figure 18. A series of aromatic amines was prepared and reacted with 4,6-dichloropyrimidine (**127**) using DIPEA in ethanol to produce **132**. This was followed by a second nucleophilic substitution reaction with various alkyl and aromatic amines, which subsequently led to products **133**.

Following optimization, lead molecule **134** expressed the highest antiproliferative potency against six different cancer cell lines, including SKOV-3–ovarian (EC_50_ = 1.5 ± 0.2 nM), HepG2–liver (EC_50_ = 1.8 ± 0.6 nM), MDA-MB-231–breast (EC_50_ = 4.4 ± 0.6 nM), HeLa–cervical (EC_50_ = 3.6 ± 0.3 nM), B16-F10–melanoma (EC_50_ = 3.3 ± 0.1 nM), and A549–lung (EC_50_ = 1.1 ± 0.2 nM). This compound also exhibited more potent antiproliferative activities than Colchicine and Paclitaxel against the paclitaxel-resistant ovarian cancer cell line A2780/T and its parental cell line A2780, indicating that **134** could overcome P-gp-mediated paclitaxel resistance in vitro. The compound also showed equal activity against lung tumors A549-WT and low EGFR expression A549, proving that EGFR inhibition was not the major reason for the antitumor activity. The PK results show that **134** can be absorbed rapidly from the intestine with t_1/2_ = 0.22 ± 0.02 h (see Table 4). The lead had a slightly high, but acceptable, CL of 69.84 ± 4.97 mL/min/kg and an AUC of 239.43 ± 16.39 ng/mL·h. These results establish that **134** has acceptable pharmacokinetic properties, and therefore, is suitable for further development.

Abdel-Aal et al. have developed anticancer compounds specifically targeting the tubulins [62]. Though anticancer drugs are already in place for this target, the authors sought to address the poor oral bioavailability and the multidrug resistance of current tubulin drugs. This work focused on the modification of the Combretastatin and Phenstatin drug scaffolds. Chalcones can be simply viewed as keto stilbenes, mimicking both Combretastatin and Phenstatin. Various modifications of the 1,3-diaryl scaffold were developed without affecting their tubulin inhibitory activity, including phenoxy substitution and replacement with heterocyclic rings [63]. The authors developed lipidated 4,6-diarylpyrimidines as tubulin polymerization inhibitors (or antiproliferative agents) which improved the interaction in the hydrophobic pocket and enhanced their physiochemical properties and cell penetration. The pyrimidine moiety in this series of compounds offered extra hydrophilic interactions and rigidity relative to the propanone scaffold, which may enhance tubulin binding.

The syntheses of molecules for this study are depicted in Figure 19. The plan targeted lipidated chalcones, which were prepared using known condensation chemistry [64]. The final lipidated 4,6-diarylpyrimidines were prepared by refluxing long-chain alkoxy-substituted chalcones **135** with urea, thiourea, or guanidine carbonate in alkaline medium to produce the required drug candidates **136**–**138**.

Eighteen chalcones and their lipidated pyrimidine derivatives were designed and synthesized as tubulin polymerization inhibitors. In general, the synthetic pyrimidine derivatives had improved antiproliferative activity over the corresponding chalcones against the MCF-7 cancer cell line. The pyrimidin-2-amine **139** showed dual antiproliferative activity against MCF-7–breast (IC_50_ = 10.95 μM) and HepG2–liver (IC_50_ = 11.93 μM) cell lines, induced apoptosis and cell cycle arrest, and displayed tubulin inhibitory activity against MCF-7 at low micromolar concentration. The compound also induced S-phase cell cycle arrest and apoptosis in MCF-7 cells with a tubulin IC_50_ = 9.7 μM. These findings established **139** as an anticancer lead worthy of further optimization and development.

A patent by Boeckman et al. describes inhibitors of histone H3K27 demethylase JMJD3 [65]. The Jumonji C (JMJC) domain, containing proteins which include histone H3K27, plays a significant role in tumorigenesis and has been identified as a key target for anti-cancer agents [66]. The patent highlights the critical role of Jumonji kinases and inhibitors of H3K27 to target diffuse intrinsic pontine glioma (DIPG), which is the most frequent brain stem tumor in pediatrics and has a survival rate of 9–12 months from diagnosis [67]. There are no surgical options for this brain stem tumor and conventional chemotherapy is used solely to alleviate pain. Due to this issue, an efficacious therapeutic agent is needed for these DIPG patients. DIPG is uniquely dependent on the H3K27 mutation for cancer initiation/maintenance and is detected in more than 80% of patients. However, more than 250 clinical trials have been executed on this target without much success.

The synthesis of prospective targets, shown in Figure 20, started with 2-cyanopyridine (**140**), which was converted to picolinimidamide hydrochloride (**141**) with ammonium chloride (NH_4_Cl) in the presence of HCl in ethanol and DCM. Hydrochloride **141** was cyclized with diethyl malonate using sodium ethoxide in ethanol to provide dihydroxypyrimidine **142,** which was transformed to the corresponding dichloro derivative **143** using POCl_3_. Subsequent S_N_Ar reaction of **143** with amine **144**, promoted by DIPEA at reflux, provided **145** [68]. Finally, **145** was reacted with various aliphatic amino alcohols to generate the desired model compounds **146**, which were evaluated for biological activity**.** Some of the final alcohols employed prodrug approaches as part of this screening.

Synthesized derivative **147** (UR-8) demonstrated selective cytotoxic activity against human DIPG-K27M cells (IC_50_ = 4–6 μM) in vitro and was apparently transported to the brain due to its in vivo stability. In a mouse study, prototype **147** showed a favorable biodistribution in the brain stem compared with a competitor compound, GSK-J1. The concentration of **147** was found to be around 4455 ± 1576 ng/mL in serum and around 409.5 ± 243.9 ng/mL in the brain stem. Extraction of brain stem tissue from mice treated with **147** followed by HPLC assay revealed 8.77 ± 2.37% of **147** in this tissue. A similar experiment using GSK-J1 (see Appendix A) detected no significant amount of the competitor compound in brain stem tissue. Further data indicate that **147** was likely active in its original form and therefore this subclass of inhibitors offers high potential for clinical application. The compound also inhibited tumor growth and prolonged survival rates in mice with human DIPG xenografts. To determine the in vivo antitumor activity of this analog, the mice were implanted with DIPG-SF8628 cells in the brain stem and treated with 100 mg/kg of **147** for 10 consecutive days. This experiment confirmed that **147** outperformed other current drugs for this tumor.

Ling et al. have developed an inhibitor for acute myeloid leukemia (AML), which is a life-threatening malignancy with a 5-year survival rate. This cancer is characterized by its disruption of hematopoietic progenitor cell differentiation and proliferation [68]. AML treatment, which includes chemotherapy, does not exhibit long-term efficacy, and 70% of people do not survive beyond 1 year [69]. Some of the known chemotherapeutic drugs, such as Cytarabine (see Appendix A) and Daunorubicin (see Appendix A), have already encountered drug resistance in patients. Bruton’s tyrosine kinase (BTK), a member of the TEC kinase family, plays a critical role in multiple signaling pathways and significantly impacts proliferation, survival, and differentiation of B-lineage and myeloid cells [70]. BTK is highly expressed and activated in more than 90% of AML patients, and, thus, could offer a potential strategy for treatment. These researchers also specified interest in a second target, namely the FMS-like tyrosine kinase 3 (FLT3), which is expressed in most AML cell lines. The authors resolved to seek a dual inhibitor for these two kinase targets to address the issue of drug resistance.

The synthesis of possible BTK/FLT3 dual inhibitors is shown in Figure 21. Two sequential S_N_Ar reactions of 2,4-dichloro-5-fluoropyrimidine (**59**), the first, at C4 by aniline esters **148** using DIPEA in isopropanol at 80 °C, gave **149**, while the second, at C2 by aniline **150** in the presence of TFA in butanol, produced **151**. Subsequently, the ester group in **151** was converted to hydroxamic acid with hydroxylamine hydrochloride and KOH in methanol to afford the final targets **152**.

Some of the compounds synthesized as BTK/FLT3 dual inhibitors exhibited IC_50_ values at low nanomolar levels. Among these dual inhibitors, **153** exhibited activity against FLT3/D835Y mutant cells with single digit nanomolar potency (IC_50_ = 5.9 ± 0.1 nM). This inhibitor showed powerful antiproliferative activity against AML cells and inhibited the growth of other leukemia cells: MV-4-11 (IC_50_ = 0.29 ± 0.02 nM) Molm13 (IC_50_ = 0.45 ± 0.03 nM), K562 (IC_50_ = 73 ± 13 nM), Molt4 (IC_50_ = 1.4 ± 0.3 nM), and THP1 (IC_50_ = 37 ± 5 nM) which are all BTK and FLT3 positive. Additionally, compound **153** effectively induced apoptosis and upregulated proapoptotic protein levels in MV-4-11 cells in a dose-dependent manner. Finally, **153** effectively suppressed the growth of MV-4-11 cells in the xenograft tumor model with a 20 mg/kg intraperitoneal (i.p.) injection and showed an antitumor effect, like Sorafenib (20 mg/kg, see Appendix A)), with no significant toxicity.

Yang et al. undertook a study to develop a drug for prostate cancer (PCa), which is a major threat to male health and results in a high mortality rate worldwide [71]. Hormonal therapies for PCa play a major role by decreasing androgen levels. However, once resistance develops in hormonal therapies, it renders this approach unusable, so there is an urgent need to develop alternative drugs for PCa [72]. In this work, the Yang group focused on dual-specificity tyrosine phosphorylation-regulated kinases (DYRKs), which belong to the CMGC kinase family, where DYRK2 plays an important role in cell proliferation, apoptosis, and migration. By downregulating DYRK2, PCa is suppressed which makes this a prominent target for inhibition [73].

The synthetic route to compounds needed for this program is shown in Figure 22. Bromobenzothiazole **154** was coupled with bis(pinacolato)diboron (**155**) in the presence of Pd(dppf)Cl_2_ and KOAc to generate boronate ester **156**, which was further coupled with pyrimidine **59** using bis(triphenylphosphine)palladium(II) chloride ((PPh_3_)_2_PdCl_2_) and K_2_CO_3_ in aqueous 1,2-dimethoxyethane (DME) to afford **157**. Compound **157** underwent Buchwald–Hartwig coupling at C4 with various protected amines **158** to provide adducts that were deprotected with ethyl acetate-HCl in DCM to deliver targets **159** for biological evaluation.

The authors used structure-based virtual screening to develop these DYRK2 inhibitors of which the most potent was **160** with an IC_50_ = 0.6 nM. This compound also elicited good inhibitory activity against proliferation and migration and promoted apoptosis on PCa cells. The ADME properties of **160** were presented along with a thermodynamic solubility of 29.5 mg/mL, a parallel artificial membrane permeation assay (PAMPA) value of log Pe = −5.98, and liver microsomal stability of ca. 16 mL/min/kg with t_1/2_ = 78 min. There was no hERG inhibition with QPloghERG = −6.743 and the compound had an excellent LD_50_ > 10,000 mg/kg. At a high concentration of 200 mg/kg, the compound displayed tumor growth inhibition better than Enzalutamide (see Appendix A), which was the positive control (100 mg/kg) in the PCa xenograft models. The mice in this study did not undergo any significant weight loss, suggesting that these compounds likely have a good safety profile.

Xie et al. have developed adenosine A_2A_ receptor (A_2A_R) antagonists as a novel strategy for cancer immunotherapy [74]. Adenosine triphosphate (ATP) is an endogenous ligand that is widely distributed throughout the human body. ATP is involved in numerous functions, including cell growth, hearth rhythm, immune function, sleep regulation and angiogenesis. Though there are four subtypes of adenosine receptor, only A_2A_R has been sufficiently investigated to attract much attention as a potential drug target for cancer and various inflammatory and neurodegenerative diseases [75]. These researchers sought to use this A_2A_R strategy to develop a treatment for colon cancer.

The synthetic route to pyrimidine derivatives for this investigation is shown in Figure 23. The synthesis leveraged two consecutive Suzuki–Miyaura couplings to 2-amino-4,6-dichloropyrimidines **161**, the first with arylboronate esters **162** to afford intermediate **163** and the second with various methyl protected pyridinones **164** to yield two sets of pyridine derivatives, **165** and **166**. Demethylation of the pyridine moieties on these structures using HBr, and subsequent *N*-alkylation afforded drug candidates **167** and **168**.

Evaluations based on SAR and ADME properties led to compound **169** with improved potency (IC_50_ = 29 nM vs. A_2A_R) and better mouse liver microsomal metabolic stability (t_1/2_ = 86 min). The compound expressed preferential activity against A_2A_R over A_1_R, A_2B_R, and A_3_R (>100-fold selectivity, IC_50_ > 3 μM), and the compound demonstrated good oral bioavailability in mice. Compound **169** showed excellent anticancer activity, with a total growth inhibition of 56.0% and good safety characteristics in the mouse MC38 colon cancer model at an oral dose of 100 mg/kg. The PK of this drug candidate was assessed in mice following i.v. (2 mg/kg) and p.o. (10 mg/kg) administration to C57BL/6 mice (n = 3 peer groups), and the results are shown in Table 5. The oral bioavailability (F) of compound **169** in mice was excellent (86.1%), and the compound had a plasma protein binding ratio of 98.6%. No significant body weight loss was observed in experimental mice, indicating that compound **169** was well tolerated at the given dosage. With these encouraging results, the anticancer agent **169** with an appended pyridinone moiety was deemed an excellent prospect for further refinement as an immunotherapeutic.

Huang et al. actuated a study to develop a hematopoietic progenitor inhibitor (HPK1) as a cancer immunotherapy [76]. HPK1 is a mitogen-activated kinase 1 (MAP4K1), a cytosolic STE20 serine/threonine kinase from the germinal kinase family which is highly expressed in immune populations, including T cells, B cells, and dendritic cells [77]. Recent evidence in this field suggests that HPK1 activation can significantly limit the intensity and duration of T-cell receptor signaling, resulting in cell dysfunction. Their results demonstrate that loss of HPK1 kinase function can increase cytokine secretion and enhance T cell signaling, virus clearance, and tumor inhibition. Thus, HPK1 has potential as a novel and effective target for cancer immune response enhancement.

In this work, rational design, synthesis, and SAR exploration were carried out for novel 2,4-disubstituted pyrimidine derivatives as potent HPK1 inhibitors by a scaffold hopping (heterocycle replacement) approach. The design of this compound was based on a reverse indazole derivative discovered by Merck, and which demonstrated highly potent and selective inhibition of HPK1 [78]. The authors used this scaffold hopping strategy for drug design and diversification of chemotypes to identify pyrimidines as alternatives for indazole rings.

Figure 24-I summarizes the preparation of one subset of target molecules. Initially, 2-chloro-4-aminopyrimidine (**170**) was condensed with 5-fluoro-2-morpholinobenzoic acid (**171**) in the presence of *O*-(7-azabenzotriazol-1-yl)-*N*,*N*,*N*′,*N*′-tetramethyluronium hexafluorophosphate (HATU) and DIPEA in THF to afford **172**. Amide **172** underwent Suzuki–Miyaura coupling with arylboronic acids/aryl(pinacolato)boronate esters in the presence of Pd(dppf)Cl_2_ and K_2_CO_3_ in dioxane to provide **173**. Access to the second subset of compounds, outlined in Figure 24-II, arose from Suzuki coupling of 2-chloropyrimidine-4-carboxylic acid (**174**) with 2-fluoro-6-methoxyphenylboronic acid (**175**) in the presence of Pd(dppf)Cl_2_ and K_2_CO_3_ in aqueous dioxane to give **176**. Finally, linkage of various anilines to **176** using HATU and DIPEA in THF furnished amides **177**.

Upon screening, the synthetic 2,4-disubstituted pyrimidines proved to be powerful and selective HPK1 inhibitors. The most promising compound, **178** (HMC-H8), potently inhibited HPK1 with an IC_50_ = 1.11 nM. The selectivity profile demonstrated that **178** exhibited good target differentiation and moderate preference against T-cell receptor-related targets such as lymphocyte-specific protein tyrosine kinase, germinal center kinase and protein kinase C-θ. In addition, the interleukin-2 (IL-2) and interferon-γ (IFN-γ) stimulation assay indicated that **178** actuated cytokine reproduction in a dose dependent manner. Notably, the reversal of immunosuppression evaluation revealed that **178** effectively restored IL-2 production, with up to 2.5 times greater increase in the IL-2 level over dimethyl sulfoxide (DMSO) treatment. The ADME properties for **178** demonstrated that the compound does not have significant CYP450 inhibition in human liver microsomes at 10 μM. The compound has low to moderate intrinsic clearance (CL_int_) = 24.37 L/min/mg in a human liver microsomal stability assay. A single PK was conducted for compound **178** on Sprague–Dawley rats (190–200 g, n = 3 peer groups) with an i.v. of 1 mg/kg and a p.o. of 10 mg/kg, and the results are summarized in Table 6. Based on the data from the table, the compound appeared to have high clearance after both i.v. and p.o. administration. Finally, the compound has a very good C_max_ and AUC with a bioavailability (F) of 15.05%.

A patent by Ding et al. synthesized a class of kinesin family member 18A (KIF18A) inhibitors specifically to treat cancer [79]. Various kinases and kinesins are responsible for division in normal cells and cancer cells. The KIF18A gene belongs to the kinesin-8 subfamily and is a plus-end oriented motor. KIF18A is thought to affect the dynamics of the plus ends of centromere microtubules to control correct chromosome positioning and spindle tension. Depletion of human KIF18A in longer spindles increases chromosome oscillations in the metaphase of HeLa cervical cancer cells and activation of the mitotic spindle assembly checkpoint. KIF18A appears to be a viable target for cancer therapy. KIF18A has been overexpressed in various cancers, including, but not limited to, colon, breast, lung, pancreatic, prostate, bladder, head and neck, cervical, and ovarian cancers. Furthermore, in cancer cells, gene deletion, knockout, or KIF18A inhibition affects the mitotic spindle body device. Inhibition of KIF18A has been found to induce mitotic cell arrest, a known weakness that can be facilitated by apoptosis, mitotic catastrophe, or heterogeneously driven lethality following mitotic slippage in interphase mitotic cell death [80].

The preparation of drug candidates for this investigation is shown in Figure 25. Initial S_N_Ar of 2-chloro-4-methyl-6-aminopyrimidine (**179**) with 4,4-difluoropiperidine (**180**) in the presence of DIPEA in 1-methyl-2-pyrrolidinone (NMP) at 140 °C produced adduct **181**. Derivative **181** was condensed with three different synthesized acids to give the final pyrimidinamide derivatives **182**. There were only three compounds reported in this patent.

The patent did not elaborate on the biological activity of the compounds but rather reported the IC_50_ values for the pyrimidinamide derivatives. The IC_50_ values for the enzymatic inhibition of KIF18A claimed for the three derivatives ranged from 27–120 nM. However, no specific data were presented.

A patent filed by Hergenrother and Kelly focused on metastatic melanoma, a cancer that readily spreads beyond its original location to other parts of the body [81]. This cancer results from genetic mutation and environmental factors. v-Raf murine sarcoma viral oncogene homolog B (BRAF) inhibitors are drugs that can shrink the growth of metastatic melanoma in patients whose tumors have a BRAF mutation. BRAF mutations are found in more than half of patients diagnosed with cutaneous melanoma. In BRAF-mutated melanoma, the BRAF kinase becomes hyperactivated, resulting in elevated cell proliferation and survival. The BRAF inhibitors Vemurafenib (**183**), Encorafenib (**184**) and Dabrafenib (**185**) are used in patients with BRAF-mutated melanoma. These inhibitors specifically target BRAF kinase and thus interfere with the mitogen-activated protein kinase signaling pathway that regulates the proliferation and survival of melanoma cells [82]. In this study, a new BRAF inhibitor, Everafenib-CO_2_H (**187**), was envisioned by combining the structural features of **183**–**185** to reduce P-gp efflux propensity as well as to enhance brain penetration and activity in challenging intracranial mouse model melanoma (Figure 4).

The synthesis of **187** is illustrated in Figure 26. In the first step, 3-amino-5-chloro-2-fluorobenzoic acid (**188**) was esterified to **189** using methanol and thionyl chloride. Ester **189** underwent reaction with propanesulfonyl chloride (**190**) using pyridine in DCM to afford the sulfonamide derivative **191** which was subsequently treated with the LiHMDS-derived anion of 2-chloro-4-methylpyrimidine (**192**) to provide **193**. Benzylic bromination of **193** and treatment with 2,2-dimethylpropanethioamide (**194**) resulted in cyclization to provide thiazole **195**. Compound **195** was then subjected to a S_N_Ar reaction with methyl 5-aminopentanoate hydrochloride (**196**) in the presence of DIPEA in *N*,*N*-dimethylacetamide (DMA) under microwave irradiation to generate **197.** Finally, hydrolysis of **197** using LiOH furnished acid **187**.

The biological properties of Everafenib–CO_2_H are summarized in Table 7. The compound displayed a similar potency in A375–human melanoma cells when compared with Dabrafenib. In cell permeability assays (Table 8), apparent permeability (P_app A-B_) is like that of Dabrafenib, but P_app B-A_ is lower, which leads to an improved efflux ratio of 1.17 ± 0.22.

Work by De Vivo et al. highlighted the targeted cell division cycle GTPases (CDC42, RHOJ, and RHOQ), which are small guanosine triphosphate (GTP)-binding proteins that are known to regulate tumor growth, angiogenesis, metastasis, and cell resistance to targeted therapies [83]. CDC42 GTPases are essential molecular switches within the cell for which their active/inactive state depends on whether they are bound to GTP or guanosine diphosphate. When CDC42 GTPases are bound to GTP, the former change their structural conformation, allowing protein surface interactions that are complementary to their downstream effectors [84]. These include, but are not limited to, p21-activated protein kinases (PAKs). Notably, PAKs are known to be involved in invasion, migration, and oncogenic transformation. Many groups have sought to design small molecules that inhibit PAK kinases by targeting the large and flexible ATP binding pocket in the kinase domain or by targeting a large auto-inhibitory region that is observed in group I PAKs (PAK1, 2, and 3). However, the developed agents have failed to reach phase 2 due to their poor selectivity. For example, existing PAK inhibitors act on multiple isoforms of PAKs, including PAK2, which is thought to induce cardiotoxicity with a narrow therapeutic window. Potential modifications to GTPase inhibitors considered by the De Vivo team are summarized on the generalized structure in Figure 5.

The synthetic plan for this work is delineated in Figure 27. In Figure 27-I, 2,4,6-trichloropyrimidine (**198**) underwent a Suzuki–Miyaura coupling with phenylboronic acid (**199**) under standard conditions to afford **200**. Subsequent S_N_Ar reaction with various substituted anilines **201** in the presence of LiHMDS in THF at −60 °C provided intermediates **202**. Suzuki coupling of **202** with **203** gave cyclic alkene **204** which was hydrogenated in the presence of ammonium formate and Pd(OH)_2_/C or triethylsilane with Pd/C to give **205**. When X = *N*-Boc, the amine **205** was deprotected with 4 M HCl in dioxane to yield **206**. In Figure 27-II, amine **207** was condensed with isobutyric acid in the presence of HATU and DIPEA in DMF to afford amide **208**. Alkylation of **207** with 1-bromo-2-methoxyethane in the presence of DIPEA in ACN at 60 °C furnished ether **209**.

Based on the recent discovery of lead compound **210**, which showed anticancer activity in vivo, the authors expanded this new chemical class of CDC42/RHOJ inhibitors. Importantly, they identified and characterized two back-up compounds, namely, **211** and **212**, derived from a SAR study with ~30 close analogs bearing different substituents on the pyrimidine or triazine core. The most potent IC_50_ values were observed from **211** against five different melanoma cell lines, including SKM28 (IC_50_ = 6.1 µM), SKMeI3 (IC_50_ = 4.6 µM), WM3248 (IC_50_ = 9.3 µM), A375 (IC_50_ = 5.1 µM), and SW480 (IC_50_ = 5.9 µM). Compound **211** also had good kinetic solubility (168 µM), a t_1/2_ > 120 min in plasma and acceptable microsomal stability (t_1/2_ = 45 min). The PK profile for compound **211** is shown in Table 9.

Back-up compounds **211** and **212** have also displayed stable binding in the target pocket via molecular dynamics simulations and favorable PK profiles comparable to **210**. Notably, the authors also measured the in vivo efficacy of the two lead compounds **211** and **212**, with analog **211** exhibiting a significant ability to inhibit tumor growth in patient-derived xenografts in vivo, similar to lead compound **210**.

Gray et al. have investigated inhibitors of the Hippo pathway, an important, evolutionarily conserved signaling cascade pathway with >30 components and which play a crucial role in organ size control, tissue homeostasis, stem cell renewal, cell proliferation, angiogenesis, and tumorigenesis [85]. Dysregulation of the Hippo pathway through merlin neurofibromin-2 loss, large tumor suppressor kinase 1 fusion, yes-associated protein (YAP) and transcriptional co-activator with PDZ-binding (TAZ) fusions, and YAP/TAZ amplification have been linked to the occurrence and progression of tumor malignancies in mesothelioma, meningioma, lung cancer, liver cancer, and other solid tumors [86]. Although the Hippo pathway has significant therapeutic potential, direct targeting of this cascade has been difficult. Thus, instead of directly targeting Hippo, the authors employed a reversible post-translational palmitoylation of the transcriptional enhanced associate domain (TEAD). Hyperactivation of TEAD–YAP/TAZ leads to human cancers and is associated with cancer cell proliferation, survival, and immune evasion. Therefore, targeting the TEAD–YAP/TAZ complex has emerged as an attractive therapeutic approach.

The synthesis of potential inhibitors for this work is shown in Figure 28-I. S_N_Ar reaction of amino ether **213** with 2-chloropyrimidine (**214**) in the presence of DIPEA in butanol at 70 °C produced amino ethers **215**. These intermediates were subsequently Boc deprotected and reacted with acryloyl chloride and TEA or condensed with acrylic acid to give amides **216**. In Figure 28-II, a 7-membered ring diamine **217** was reacted with **214** using DIPEA in DMSO at 90 °C to provide **218**. Intermediate **218** underwent the same amide formation with acryloyl chloride to afford target **219**.

Time-resolved fluorescence energy transfer and TEAD reporter assays in this work demonstrated that the overall Y-shaped scaffold improved the potency of the compounds to an IC_50_ < 50 nM. The results suggest that selectivity could be achieved between TEAD isoforms due to modifications in different parts of the ring. Optimization of the chemistry on this series of compounds resulted in the development of a potent pan-TEAD inhibitor **220** (MYF-03-176). This structure exhibited potent inhibition of TEAD transcription with an IC_50_ = 17 ± 5 nM and significantly inhibited TEAD-regulated gene expression and proliferation of the cell lines with TEAD dependence, including those derived from mesothelioma and liposarcoma. Compound **220** also expressed the best antiproliferation activity on both the 94T778–liposarcoma (IC_50_ = 40 nM) and NCI-H226–squamous cell carcinoma (IC_50_ = 24 nM) cell lines.

A patent invention by Yu et al. sought to use small molecule pyrimidine derivatives to mitigate proliferative disorders caused by the expression of various kinases [87]. The required compounds would need to inhibit the growth of wild tumor beads with high kinase expression and tumor cell lines with corresponding kinase mutations. These proliferative disorders include rearranged transfection (RET), glial-derived neurotrophic factor (GDNF), platelet-derived growth factor receptor (PDGFR), and vascular endothelial growth factor receptor (VEGFR). Kinase-derived RET is a neuronal growth factor receptor tyrosine kinase and a transmembrane glycoprotein. The proto-oncogene, located on chromosome 10, is expressed during the embryonic stage, plays an important role in the development of the kidney and enteric nervous system, and is also critical in the homeostasis of neurons, neuroendocrine cells, hematopoietic tissue, and male germ cells [88]. These RET kinase inhibitors may find use in the treatment of cancer and gastrointestinal disorders. The growth of solid tumors is highly dependent on vascular proliferation, especially PDGFR and VEGFR. These are the main mediators of angiogenesis and act as two indicators of the angiogenic potential of human gliomas. Neurturin and persephin are ligands belonging to the GNDF family (GFLs). GFLs usually bind to the GDNF family receptor α (GFRa), and the formed GFL–GFRa complexes mediate the self-dimerization of RET proteins, causing trans-autophosphorylation of tyrosine in the intracellular domain. This complex also recruits related adapter proteins and activates the cascade reaction of signal transduction such as cell proliferation and related signaling pathways that include mitogen-activated protein kinase (MAPK), phosphoinositide 3-kinase (PI3K), Janus kinase signal transducer and activator of transcription (JAK-STAT), protein kinase A (PKA), and protein kinase C (PKC). Thus, the patent sought a small molecule inhibitor to block these kinases in order to restrict the proliferation of cancer cells.

A route to small molecule kinase inhibitors for this project is outlined in Figure 29. Sandmeyer reaction of 2-amino-4-chloropyrimidine derivatives **221** using tert-butylnitrite and diiodomethane in ACN solvent afforded iodopyrimidines **222**. Intermediates **222** underwent a S_N_Ar reaction with various amines and DIPEA to form **223**. Anilinic amines **223** were Boc-protected to give **224** and subjected to Heck coupling with methyl acrylate using Pd(OAc)_2_ and TEA in ACN to give **225**. Acrylic esters **225** were hydrolyzed to acid **226** using LiOH and condensed with another set of amines using standard conditions to give amides **227**. Final Boc deprotection of **227** with TFA in DCM delivered the required drug candidates **228**.

Compounds **228** were evaluated as inhibitors of WT and mutant RET kinases, namely RET-V804M and RET-V804L. Several compounds recorded potencies ranging from 1 nM–1 μM in these assays. Compound **229** exhibited the best activity on two cell lines, TT–human thyroid and KM12–human colon adenocarcinoma, with IC_50_ values of 22 nM and 1.16 nM, respectively. Prototype **229** also showed inhibitory activity toward KIF5B-RET fusion (IC_50_ = 22 nM) and CCDC6-RET fusion cell lines (no IC_50_ given) which were developed to establish the antitumor activity. Most compounds tested on these cell lines showed IC_50_ values between 20 nM and 1 μM. The antitumor activity was determined by a pharmacodynamic model of human cancer in BALB/c nude mice with a xenograft tumor derived from the TT cell line. Compound **229** gave tumor shrinkage of close to 80% at a dose of 40 mg/kg thereby exhibiting a robust anti-tumor effect. Finally, the mice showed no significant change in body weight which signified a good tolerance for the compound.

Chen et al. reported the synthesis of inhibitors toward salt-inducible kinases (SIKs), intracellular serine/threonine kinases which belong to the adenosine monophosphate activated (AMPK) superfamily [89]. The important role of SIKs is to act as molecular switches to regulate the transformation of macrophages (M1/M2) by phosphorylating CREB regulated transcription co-activator 3 (CRTC3) and to control its localization by activating the CRTC3 gene [90]. Some of the SIKs are involved in tumor cell resistance to cell-mediated immune responses and in resistance to tumor necrosis factor. In this work, the authors were focused on improving the ADME and pharmacokinetic properties of a known SIK inhibitor, HG-9-91-01 (see Appendix A), which suffered from poor drug properties, including rapid clearance, low in vivo exposure, and high plasma protein binding [91]. To overcome these deficiencies, the authors sought to hybridize Dasatinib (see Appendix A) and HG-9-91-01 to optimize the drug properties.

The syntheses of prospective drug molecules for this study are shown in Figure 30. The starting 2,4-dichloropyrimidine ester **230** underwent C4 substitution with various alkyl amines in the presence of TEA in ACN to afford **231** which was subsequently hydrolyzed with LiOH in aqueous THF to form **232**. Acids **232** were further transformed to their acid chlorides which reacted with substituted benzylamines and 2,6-dimethylaniline to provide amides **233** and **235**, respectively. Finally, these compounds underwent C2 S_N_Ar reaction with a series of anilines in AcOH to furnish the desired derivatives **234** and **236**.

Once these compounds were available, the pharmacokinetic profiles were evaluated. Each compound showed a modest improvement with a longer half-life, lower clearance and enhanced metabolic stability to human liver microsomes (t_1/2_ = 120 min). The plasma protein binding for the most promising structure, **237**, was ca. 79.4%, compared with >99% for HG-9-91-01. In addition to demonstrating good SIK inhibitory activity, **237** had medium selectivity among the subtypes of SIKs and exhibited excellent anti-inflammatory properties in a dextran sulfate sodium-induced colitis model. The in vitro anti-inflammation activity evaluation by cell-based phenotypes for **237** showed SIK inhibition via up-regulated IL-10 and reduced IL-12 at both the gene and protein level. The macrophage markers were also observed in LIGHT, SPHK1 and Arginase 1 proteins for the best compound synthesized. The PK profile of **237** is condensed in Table 10.

A patent disclosure from Marseglia et al. also imagined inhibitors targeting SIK3 [92]. SIK3 is involved in tumor cell resistance to cell-mediated immune responses, specifically, tumor cell resistance to tumor necrosis factor. Recent reports have demonstrated that SIK3 expression regulates transforming growth factor-β mediated transcriptional activity and apoptosis [93].

The synthesis of a drug candidate for this study is shown in Figure 31. S_N_Ar reaction of 4,6-dichloro-2-methylpyrimidine (**17**) by ethyl 2-aminothiazole-5-carboxylate (**238**) in the presence Cs_2_CO_3_ in DMF furnished aminothiazole-pyrimidine **239**. A second S_N_Ar reaction of **239** with 4-methylpiperazine using DIPEA in butanol produced **240**, which was saponified to acid **241**. Amidification of **241** with 3-aminothiophene in the presence of chloro-*N*,*N*,*N*′,*N*′-tetramethylformamidinium hexafluorophosphate (TCFH) and DIPEA in ACN generated amide **243**.

Model compound **243** inhibited both SIK2 and SIK3 with IC_50_ values of ca. 10 and 20 nM, respectively. The compound also exhibited significant inhibition of (1) nuclear factor kappa-light chain enhancer of activated B cells (NF-κB) in MC38–colon and EMT6–epithelial carcinoma cells and (2) histone deacetylase 4 phosphorylation. Agent **243** had a reasonable ADME profile, with a human liver microsomal stability of 131 μL/min/mg and a human hepatocyte stability of 58 μL/min/mg. Compound **243** also demonstrated good plasma exposure with 1% unbound and a recovery of 92% after 4 h. Moreover, **243** showed good permeability with Madin Darby canine kidney MDR1 cells with a P_app A>B_ of ca. 5 × 10^−6^ cm/s. However, the efflux ratio was nearly 26, suggesting that the compound was an efflux substrate. Compound **243** presented a good PK profile, exhibiting low clearance with a good AUC following p.o. administration of 30 mg/kg (Table 11).

Finally, treatment of established tumors (MC38–colon, EMT6–epithelial) in different syngeneic tumor mouse models with **243** resulted in significant tumor growth inhibition in a monotherapy protocol. Compound **243** showed tumor shrinkage of 74% with a 25 mg/kg twice daily dosing and a body weight increase of 16.8%, which was comparable or even superior to anti-programmed cell death-1 treatment alone. Furthermore, immune cell profiling of treated mice showed a significant infiltration of activated T cells, along with excellent reduction in immunosuppressive regulatory T cells and M2 tumor-associated microphages.

## 4. Pyrimidine-Based Drugs for Immunological Treatments

A patent by the Zhaoxing team investigated Janus kinase 2 (JAK2), which is an important pathogenic factor for various diseases [94]. Upon activation of the JAK kinases by receptor activators, these enzymes phosphorylate cytokine receptors and activate the signal transducers and activators of transcription (STAT) family [95]. In recent years, the therapeutic potential of JAK inhibitors has focused on diseases affecting various pathological conditions of the immune system, including atopy, cell-mediated hypersensitivity (allergic contact dermatitis, hypersensitivity pneumonitis), systemic lupus erythematosus, rheumatoid arthritis, psoriasis, transplantation (graft rejection, graft-versus-host disease), etc. [96]. Recently, erythropoietin-JAK2 signaling pathways have been implicated in myeloid proliferative disorders and proliferative diabetes mellitus, which are important in omental disease. Fedratinib (see Appendix A), a JAK2 inhibitor, has been approved by the U.S. and is currently on the market. There are multiple deficiencies in this inhibitor, including its degree of selectivity for JAK2. The low selectivity, low bioavailability, and high toxicity limits its safe drug use in clinical practice. Therefore, it is necessary to develop a new more selective inhibitor of JAK2 that can overcome these shortcomings.

The synthesis of a prospective JAK2 inhibitor is shown in Figure 32. Treatment of 2,4-dichloro-5-trifluoromethylpyrimidine (**105**) with aqueous ammonia in THF afforded the corresponding aminopyrimidine **244**. Intermediate **244** underwent Buchwald–Hartwig coupling with 4-bromobenzenesulfonamide **245** using Pd_2_(dba)_3_, xantphos, and Cs_2_CO_3_ in dioxane to afford the corresponding sulfonamide-pyrimidine derivative **246**. Derivative **246** was further subjected to a S_N_Ar amination reaction with the 4-substituted aniline **247** to generate drug candidate **248**. Following conversion to various salts, this compound was evaluated for its biological significance against the JAK2 kinase.

Compound **248** or its salts had a 96-fold preference for JAK2 over JAK1, whereas the Fedratinib competitor showed only a 3-fold preference. Prototype **248** was also very selective for JAK2 kinase with an IC_50_ = 5.86 nM and was much less active toward JAK3 (IC_50_ = 538.5 nM) and tyrosine kinase 2 (TYK2, IC_50_ = 700.4 nM).

A PK study was conducted for several of the compounds prepared, and the results for the most potent, **249** (the fumarate salt of **248**), when administered p.o. (10 mg/kg), are shown in Table 12. The compound showed a very low clearance, and the oral bioavailability (F) of the salt was 13.4%, whereas the Fedratinib competitor was 7.24%. In a guinea pig allergic conjunctivitis model, however, it was found that **249**, even at a 10-fold lower dose, elicited a better therapeutic effect than Fedratinib.

An investigation by Ellis and co-workers focused on developing a drug scaffold aimed at dry eye disease (DED) which affects more than 39 million adults in the US [97]. To date, only four therapies for this affliction have been approved by the FDA. Though the target and mechanism are not well defined, it is understood that there are underlying cytokine and receptor-mediated pathogenic inflammatory states conspiring to break T cell and B cell tolerance against self-antigens, resulting in an undesirable autoimmune response that leads to ocular surface inflammation and loss of tear film homeostasis [98]. The work in this article focused on the Janus family of intracellular tyrosine kinases. Since many of the cytokines signaling through the JAKs (IL-2 (JAK1/JAK3), IL-6 (JAK1/JAK2/TYK2), IL-12 (JAK2/TYK2), IL-23 (JAK2/TYK2), and IFN-γ (JAK1/JAK2)) are implicated in the immunoinflammatory pathogenesis and pathophysiology of DED [99], a new small molecule JAK inhibitor, which is potent and water-soluble, could represent an ideal drug for pharmacological intervention against this condition.

The synthesis of several potential anti-inflammatory agents is summarized in Figure 33. In Figure 33-I, 2,4-dichloro-5-methylpyrimidine (**250**) underwent a regioselective Suzuki−Miyaura cross-coupling with (*R*)-3-cyclopentyl-3-(4-(4,4,5,5-tetramethyl-1,3,2-dioxaborolan-2-yl)-1*H*-pyrazol-1-yl)propanenitrile (**251**) to provide **252** using (Ph_3_P)_4_Pd and Na_2_CO_3_ in aqueous dioxane at 100 °C. Subsequent linking of **252** with 1-methyl-1*H*-pyrazol-4-amine (**253**) under Buchwald–Hartwig conditions using (Ph_3_P)_4_Pd, 2,2′-bis(diphenylphosphino)-1,1′-binaphthalene (BINAP) and K_2_CO_3_ in dioxane–*t*-butanol at 100 °C completed the synthesis of target **254**. In Figure 33-II, a similar Suzuki–Miyaura coupling was carried out between **255** and **256** to afford **257**, which underwent *m*-chloroperoxybenzoic acid (*m*-CPBA) oxidation with sulfur to give sulfone **258**. Independent S_N_Ar displacement of the sulfone in **258** with two amines yielded products **259a** and **259b**. In Figure 33-III, S_N_Ar displacement of the C4 chloride of 2,4-dichloro-5-methylpyrimidine (**250**) with tert-butyl 3,9-diazabicyclo[3.3.1]nonane-9-carboxylate (**260**) using DIPEA in DMF gave **261**. Intermediate **261** was then coupled with amine **253** under Buchwald–Hartwig conditions and Boc deprotected with ethanolic HCl to generate **262**. Final *N*-alkylation of **262** with either bromoacetonitrile or acrylonitrile in the presence of DIPEA delivered target **263**. In Figure 33-IV, regioselective S_N_Ar displacement of the C4 chloride from **250** with 2-(azetidine-3-yl)acetonitrile hydrochloride (**264**) using DIPEA in DMF afforded **265**. Subsequent conjugate addition of 1-methylpiperazine to this intermediate in the presence of DBU yielded **266**. Finally, Buchwald–Hartwig amination of **266** with 2-methylisothiazol-3-amine (**267**) completed the synthesis of prototype **268**.

A pharmacophore-based SAR helped to identify a lead JAK inhibitor as an immunomodulating anti-inflammatory agent for topical ocular disposition. Compound **268**, which had a unique 3-aminoazetidine bridging scaffold, offered good JAK-STAT potency and excellent aqueous solubility. Overall, **268** displayed suitable, low single-digit nanomolar potency toward JAK2 (IC_50_ = 3.9 nM) and good on-target cellular potency (STAT3 (IL-6), IC_50_ = 162 nM), excellent aqueous solubility (24,904 μM), minimal off-target kinase activity (S(35) = 0.055), and no observable genotoxicity in the micronucleus (a biomarker for genotoxicity) assay at the highest concentration tested (3 × micronucleus concentration, ≥50 μM). The pharmacodynamics of **268** were evaluated on a murine model of allergic eye disease in vivo against inflammation-driven Meibomian gland dysfunction. A three-day study using 0.1% and 0.3% of **268** demonstrated marked improvement in clinical scores relative to the vehicle control. Notably, no statistical differences were observed at any timepoint between 0.3% **268** and 0.1% Dexamethasone (positive control, see Appendix A)). This striking result strongly indicates the potential of **268** to affect a rapid, sustained, and robust anti-inflammatory response by inhibition of the JAK-STAT pathway. Thus, compound **268** was deemed a safe, fast-acting and well-tolerated noncorticosteroid eye drop to treat DED as well as other inflammatory ocular surface diseases.

A patent by Zhang et al. pursued the development of 2,4-diarylaminopyrimidine derivatives for treatment of inflammation [100]. The authors did not specify the mode of action or the target, though the work emphasized the use of lipopolysaccharides (LPS) to stimulate the inhibitory activity of human airway epithelial cells to release inflammatory cytokines. The LPS-induced human airway epithelial cell inflammation model was used to evaluate the anti-inflammatory activity of the synthesized derivatives [101].

The synthesis of potential anti-inflammatory candidates is summarized in Figure 34. In this work, various aliphatic and aryl amines participated in a S_N_Ar reaction to displace the C4 chloride in 2,4-dichloropyrimidine (**52**) in the presence of TEA in ethanol or DIPEA in *t*-butanol to provide **269**. Subsequent reaction of derivatives **269** with various nitroaryl amines **270**, promoted by *p*-TsOH·H_2_O in dioxane, gave nitroaryl pyrimidine-diamines **271**, which upon reduction, gave aminoaryl pyrimidinediamines **272**.

All of the prepared derivatives expressed good inhibitory effects on inflammatory cytokines IL-6 and IL-8. The effective compounds also had inhibitory effects on the release of IL-6 and IL-8 stimulated by LPS. The authors have asserted that all of the compounds in this series demonstrated excellent inflammatory response and showed high bioavailability. Among the series evaluated, the patent claimed that compound **273** exhibited the best anti-inflammatory activity at 5 μM concentration with inhibition of IL-6 and IL-8 reaching 66% and 71%, respectively. Several of the candidate compounds were endorsed as having high potential as future drugs.

## 5. Pyrimidine-Based Drugs for the Treatment of Neurological Disorders

A patent by the Defossa team focused on inflammatory responses to harmful stimuli, such as the invasion of pathogens and tissue damage [102]. Chronic inflammation is an important underlying factor in many human diseases, such as neurodegeneration, rheumatoid arthritis, autoimmune and inflammatory diseases, and cancer. Receptor-interacting protein kinase 1 (RIPK1, UniProtKB Q13546) is a key regulator of inflammation, apoptosis, and necroptosis. Receptor-interacting protein kinase 1 has an important role in modulating inflammatory responses mediated by NF-κΒ [103]. Dysregulation of RIPK1 signaling can lead to excessive inflammation or cell death, and, conversely, inhibition of RIPK1 can be an effective therapy for chronic neurodegenerative diseases involving inflammation or cell death. RIPK1 inhibition has been identified as a promising target for different diseases, like rheumatoid arthritis, psoriasis, multiple sclerosis, and Alzheimer’s disease, and of inflammatory bowel diseases, such as Crohn’s disease or ulcerative colitis [104]. Dihydropyrazoles and isoxazolidines as RIPK1 inhibitors are well known in phase II clinical trials for ulcerative colitis [105].

The syntheses of prospective agents to treat these conditions are shown in Figure 35. Reaction of 4,6-dichloropyrimidine (**127**) with the 4-piperidinecarboxylic ester hydrochloride **274** in the presence of DIPEA in butanol afforded C4-aminated intermediate **275**. S_N_Ar reaction of **275** at C6 with imidazoles **276** and triazoles **278** using K_2_CO_3_ and Cs_2_CO_3_ in various solvents afforded **277** and **279**. Compounds **279** were further transformed by saponification of the 4-piperidinyl ester to give **280** and condensation with the 3-arylisoxazoline derivative **281** to give **282**. Incorporation of a pyrazole on the pyrimidine ring of **275** was achieved by Suzuki–Miyaura coupling with **283** to generate **284**. On the other hand, a methyltriazole was added by reaction of **275** with hydrazine hydrate in isopropanol, followed by condensation with *N*-(*E*)-[(dimethylamino)methylidene]acetamide (**285**) using catalytic *p*-TsOH·H_2_O in ethanol to afford **286**.

The compounds were evaluated in the receptor-interacting protein kinase 1 (RIPK1) inhibition assay. The catalytic activity of RIPK1 was measured using an ADP-Glo Kinase Kit and the cellular assay was measured in U937–lymphoma cells for RIPK1 inhibition activity causing cell death. The patent spotlighted 20 compounds in both assays and the potency for most compounds ranged from 4–200 nM. Overall, compound **287** performed the best, having an IC_50_ = 9 nM in the RIPK1 kinase inhibition assay and an IC_50_ = 4 nM in the U937 cellular assay.

Hartz et al. sought to develop a drug for Alzheimer’s disease, which is a neurodegenerative disorder characterized by memory loss and cognitive impairment [106]. As the disease progresses, it also causes deterioration in behavioral functions leading to communication problems, spatial disorientation, and changes in personality. More than 5.8 million Americans over the age of 65 currently live with Alzheimer’s disease [107]. In this work, the authors targeted glucogen synthase kinase-3 (GSK-3) which is a proline-directed serine/threonine kinase that is widely distributed in the human body. GSK-3β is an important isoform of GSK-3 found in most areas of the brain. The most recent studies suggest that this kinase offers a therapeutic window wherein the GSK-3β inhibition that modulates key neuronal molecular targets can be achieved while avoiding mechanism-based β-catenin-driven effects [108]. As a result of its multifaceted role, GSK-3 has been linked to numerous conditions, including Alzheimer’s disease, mood disorders and type 2 diabetes as well as cancer and myocardial disease.

The syntheses of candidate structures for this research are shown in Figure 36. In Figure 36-I, 2-aminopyrimidine-4-carboxylic acid (**288**) was reacted with a set of 3-aminopyridines **289** in the presence of HATU with DIPEA in DCM or DMF to afford amides **290**. One of the compounds prepared, **291**, was further subjected to a Buchwald–Hartwig coupling with bromobenzene (**292**) in the presence of Pd_2_(dba)_3_, xantphos, and Cs_2_CO_3_ in dioxane to provide targets **293**. Figure 36-II employed the same reaction conditions as above, starting with 2-chloropyrimidine-4-carboxylic acid (**294**) along with a different subset of 3-aminopyridines **295,** to provide amides **296**. Compounds **296** underwent amination either by a Buchwald–Hartwig coupling or by S_N_Ar reaction in NMP at 150 °C to furnish prototypes **297**.

In this investigation, two highly potent pyrimidine-based GSK-3 inhibitors were discovered. Amides **298** and **299** displayed potent IC_50_ values of 0.35 nM and 0.56 nM on GSK-3β and inhibition of p-tau with IC_50_ values of 10 nM and 34 nM, respectively. However, both compounds also exhibited respective IC_50_ values of 0.25 nM and 0.45 nM on GSK-3α, as this isoform has a 90% similarity to GSK-3β. Some of the ADME properties of **298** and **299** are summarized in Table 13. PAMPA and Caco-2 assay results show that these compounds are highly permeable to membranes. Kinase selectivity assessment in the Ambit panel of 412 kinases indicated that analogs with an aromatic group at the 4-position of the pyridine exhibited excellent kinase discrimination. Both **298** and **299** showed high selectivity against CDK2 and CDK5 in a standard panel of kinases.

The PK properties of **298** and **299** are summarized in Table 14. Both compounds had excellent oral bioavailability in a triple-transgenic mouse Alzheimer’s disease model with a good volume of distribution. The compound had low clearance and an average t_1/2_. In vivo studies have demonstrated that these compounds were brain-penetrant GSK-3 inhibitors that significantly lowered tau phosphorylation. The results described herein may encourage further investigation of this class of GSK-3 inhibitors as a potential treatment for Alzheimer’s disease.

A recent patent disclosure by Kumaravel et al. describes the utilization of pyrimidine-based compounds to treat a wide variety of neurological disorders attributable to transactive response DNA binding protein-43 (TDP-43), such as immunoreactive pathology, chronic traumatic encephalopathy, amyotrophic lateral sclerosis (ALS), Parkinson’s disease, Alzheimer’s disease, myofibrillar myopathy, sporadic inclusion body myositis, dementia pugilistica, chronic traumatic encephalopathy, Alexander disease, progressive supranuclear palsy, corticobasal degeneration, and frontotemporal lobar degeneration [109]. TDP-43 is an important nuclear DNA/RNA binding protein involved in RNA splicing. With pathological stress, TDP-43 translocates to the cytoplasm and aggregates into stress granules and related protein inclusions [110]. These phenotypes are known to degrade motor neurons and are found in 97% of all ALS cases. These TDP-43 mutations promote aggregation and are linked to a higher risk of developing ALS, suggesting that protein misfolding and aggregation act as drivers of toxicity [111]. In this invention, inhibitors of a Fyve-type zinc finger containing phosphoinositide kinase (PIKfyve) were prepared and studied in order to treat or prevent neurological disorders, such as the conditions enumerated above. The disclosure was based, in part, on the discovery that PIKfyve inhibition modulates TDP-43 aggregation in cells. Suppressing this aggregation exerts beneficial effects in patients suffering from neurological decline.

The synthetic work from this patent is outlined in Figure 37 and involves the construction of the pyrimidine core appended with a morpholine. Reaction of morpholine-4-carboximidamide hydrochloride (**300**) with 2-methoxymalonate ester **301** in the presence of methanolic NaOMe provided **302**. Dihydroxypyrimidine **302** was reacted with POCl_3_ to give the dichloride **303** which served as the starting material for a variety of analogs. Initially, **303** was subjected to Suzuki coupling with boronate ester **304** with Pd(dppf)Cl_2_ and Cs_2_CO_3_ to generate **305**. Subsequently, **305** underwent methoxycarbonylation with CO and methanol in the presence of Pd(OAc)_2_, Pd(dppf)Cl_2_, and TEA to give ester **306**. This ester was converted to an acid which was condensed with amines using HATU to yield **307**. Chloropyrimidine **305** was also subjected to Buchwald–Hartwig coupling with a series of amines to afford targets **308**. A similar approach, in which intermediate **303** was reacted with various alkyl alcohols in the presence of NaH gave **309**. Final S_N_Ar reaction with a series of amines resulted in **310**.

Some of the analogs synthesized in this patent were evaluated in a biochemical inhibition assay for PIKfyve, and the results reveal more than 10 compounds exhibiting single-digit nM potency. The compounds were further evaluated in a PIKfyve early endosome antigen 1 assay in which the best compounds demonstrated activity in the 10–100 nM range. The compounds were further evaluated with ferrous amyloid buthionine 1 (FAB1) mouse and PIKfyve TDP-43 yeast models. Compounds that demonstrated low nanomolar potency in the biochemical assay were also active in the PIKfyve TDP-43 yeast model while structures that showed weak activity in the biochemical assay proved ineffective in the PIKfyve TDP-43 models.

An invention publication by Lei et al. explored two small molecule inhibitors of Parkinson’s disease [112]. Previous research by others [113] has revealed that leucine-rich repeat kinase 2 (LRRK2) might prove to be a key link to understanding the etiology of Parkinson’s disease. The LRRK2 mechanism functions by blocking the molecular chaperone mediation which induces autophagy and leads to the degradation of α-synuclei that results in toxicity. This chemical mechanism also induces mitochondrial damage and endolysosomal dysfunction, which paves the way for disease progression. LRRK2 kinase inhibitors can reduce damage in a Parkinson’s disease model and can improve motor function in patients. Nitrogen-containing heterocycles, especially the pyrimidine core skeleton, are known to play an important role as LRRK2 inhibitors. This invention aimed to create a drug that could effectively inhibit LRRK2 as a potential treatment for neurodegenerative diseases. The patent identifies several drug scaffolds that are brain penetrating and incorporates these in effective inhibitors of LRRK2 as well as mutant LRRK2 that could be used as a treatment option for chronic neurodegenerative diseases.

In this patent, the requisite pyrimidine derivatives were assembled in two steps shown in Figure 38. Reaction of 2,4-dichloro-5-(trifluoromethyl)pyrimidine (**105**) with methylamine hydrochloride in methanol afforded **311**. Intermediate **311** was then independently subjected to S_N_Ar reaction with aminopyrazole-piperidine derivatives **312** and **314** in the presence of 0.5 N HCl in *t*-butanol at 80 °C to produce candidate LRRK2 inhibitors **313** and **315**.

While little about the biological properties of these compounds is revealed in this patent, the compounds synthesized are claimed to have IC_50_ values of ca. 13.0 nM compared with a known LRRK2 inhibitor which had an IC_50_ = 90 nM. Though the patent reports that **311** and **313** have better pharmacological profiles than those of known LRRK2 inhibitors, no data are included to support this claim.

A paper by Gelin et al. aimed to develop a potent brain penetrant and a selective glutamate receptor N2B (GluN2B) inhibitor [114]. Glutamate receptors serve an important function in neuronal activity, by regulating the brain’s predominant excitatory neurotransmitter. Dysfunction of the glutamate receptor *N*-methyl-D-aspartate (NMDA) leads to many neurological and psychiatric disorders, including Alzheimer’s disease, Parkinson’s disease, neuropathic pain, stroke, brain trauma, schizophrenia, and depression [115]. It has been shown that modulation of NMDA with a ketamine antagonist results in very robust antidepressant activity.

The preparations of several pyrimidine–triazoles for this investigation are shown in Figure 39. A flow chemistry method was used to convert various aniline derivatives to aryl azides that were cyclized with propargyl alcohol to produce 1,2,3-triazole alcohols **316**. Deprotonation of these alcohols with NaH in DMF and S_N_Ar etherification of 2-chloropyrimidine **317** delivered the target derivatives **318**.

The pyrimidine–triazole ethers were evaluated and shown to exhibit very favorable profiles, especially with respect to cardiovascular safety issues. Optimization of both the potency and metabolic characteristics of these model compounds was achieved by the introduction of a metabolic soft spot (a C4-methoxymethyl on the pyrimidine) to trigger metabolic switching. This design feature in the most active compound **319** precluded the formation of metabolites M1 and M2, a result of the loss of the pyrimidine moiety’s ability to afford the triazole alcohol and acid, which in turn manifested in a saturable, nonlinear PK. Some of the PK parameters are condensed for **319** in Table 15.

Diether **319** also showed very high aqueous solubility, and the compound was highly selective for GluN2B negative allosteric modulator (hGluN2A/C/D, IC_50_ > 10 μM) over other isoforms. Compound **319** did not have any hERG drawbacks or drug–drug interactions (over 10 μM) and the compound was not found to be a P-glycoprotein (P-gp) substrate. Candidate **319** also achieved 77% GluN2B receptor occupancy 0.5 h after a p.o. dose of 10 mg/kg, with excellent brain permeation (unbound partitioning coefficient, K_p,uu_ = 0.65). Finally, the compound also had an efficacious plasma EC_50_ = 541 ng/mL and brain EC_50_ = 121 ng/mL.

A patent invention by Wagner, et al. sought to develop novel small molecule splicing modulators (SMSMs) for use in treating a variety of diseases, including neurodegenerative and repeat expansion diseases [116]. The disclosure primarily focused on the neurodegenerative disorder known as Huntington’s disease by targeting the spliceosome. Currently, there is no cure for Huntington’s disease or any way to mitigate its progression. Splicing is carried out by spliceosomes and is an essential process for generating distinct transcripts in different cells and tissue types during the developmental process [117]. Most cases of the disease are caused by mutation in the spliceosome, while others arise from mutations at the splicing sites, branchpoints, or by various splicing enhancers and silencers. Small molecules, such as RNA splicing modulators, are a recent area of exploration for identifying small molecule modulators with limited chemical series. Thus, there is a great need in this area for the discovery of SMSMs, due to the ability of small molecules to be effective delivery options with good bioavailability.

Access to potential SMSMs for this research is outlined in Figure 40. Starting with thioether-substituted 4-chloropyrimidine ester **320**, etherification by S_N_Ar displacement of the C4 chloride afforded **321**. Compounds **321** were converted to **322** by oxidation of the thioethers to the sulfones with *m*-CPBA. The sulfone groups were displaced by various amines in the presence of K_2_CO_3_ in ACN to provide the 2-aminopyrimidine esters **323**. Hydrolysis of the ester function in **321** with LiOH gave acids **324** which were converted to amide products **325**.

The patent did not divulge any ADME or PK properties of the drug candidates synthesized but only claimed IC_50_ values < 500 nM for some of the promising compounds on the minigene reporter assay PMS1.

Another patent disclosure by Burli and Doyle promoted the invention of *N*-(4-aminocyclohexyl)pyrimidine-4-carboxamides as brain permeable cluster of differentiation 38 (CD38) inhibitors for treating disorders associated with CD38 activity [118]. Nicotinamide adenine dinucleotide (NAD+) is an essential cellular component in most living organisms and is responsible for redox functions. Though this is the primary role of NAD+ in most organisms, there are other functions for which NAD+ is important. An example of this is the necessity for NAD+ to be maintained to ensure long-term tissue homeostasis. Due to aging, there is a decrease in NAD+ levels, which lowers metabolic function [119] and leads to debilitating conditions, such as Alzheimer’s and Parkinson’s disease. One way to stop the consumption of NAD+ is by inhibiting CD38, which has emerged as a valuable therapeutic approach for age-related disorders. CD38 is a multifunctional protein involved in (1) cellular NAD+ homeostasis via its hydrolase function and (2) the generation of second messengers such as adenosine diphosphate ribose (ADPR) and cyclic-ADPR. Several experiments using CD38 knockout mice have demonstrated the positive effects of CD38 deletion in models of neurodegeneration.

The synthesis of several potential pyrimidine CD38 inhibitors was accomplished in concise fashion as depicted in Figure 41. Compound assembly was initiated by reacting 2-chloropyrimidine ester **326** with various five-membered nitrogen heterocycles **327** in the presence of DIPEA or Cs_2_CO_3_ in DMF at 100 °C to afford **328**. Hydrolysis of these heterocyclic pyrimidine esters **328** with LiOH in THF afforded acids **329**, which were reacted with various amines in the presence of TEA and propylphosphoric anhydride (T_3_P) to deliver the required amides **330**.

The analogs synthesized were evaluated for their CD38 hydrolase activity. Many of the compounds showed strong CD38 inhibition at ca. 40 nM or lower, including candidate **331**. In terms of pharmacokinetics, the tissue binding assay revealed a wide percentage range of unbound compound in mouse brain. Several compounds showed between 63–68% of unbound compound in mouse brain, with 12–22% of unbound compound in mouse plasma. A single PK study using 10 mg/kg p.o. was carried out to access the PK in brain permeability and revealed that **331** was the most promising derivative (see Table 16). The results demonstrate a robust 5512 nM concentration of **331** in the brain with a free brain concentration around 3801 nM and an unbound partitioning coefficient (K_p,uu_) of around 1.33. The *N*-(4-aminocyclohexyl)pyrimidine-4-carboxamides displayed excellent brain permeability, whereas the corresponding cyclohexyl ethers or alcohols showed very low brain concentrations and lower K_p,uu_ values.

## 6. Pyrimidine-Based Drugs for the Treatment of Chronic Pain

A patent from Eli-Lilly focused on developing a potentiator for the human mas-related G-protein coupled receptor member X1 (hMRGXI) to address the issue of chronic pain [120]. This condition is often associated with older adults due to restricted mobility in daily activities. The major problem when treating chronic pain is due to dose-limiting adverse reactions, such as addiction, which is a problem for many analgesics currently available on the market. In this patent, several 2-aryloxy- and 2-arylthio-substituted pyrimidines that act as antagonists against the corticotropin releasing factor receptor were advanced to treat conditions such as depression, anxiety, drug addiction, and inflammatory disorders [121]. In this disclosure, certain (trifluoromethyl)pyrimidine-2-amines were identified as potentiators of hMRGXI that might prove to be a viable means to solve the issue of chronic pain.

The synthetic work from this patent is summarized in Figure 42. Ethyl 4,4,4-trifluoro-3-oxobutanoate (**332**) reacted with sodium hydride and iodomethane-d_3_ in methyl tert-butyl ether (MTBE) under reflux to afford **333**. Derivative **333** further underwent cyclization with guanidine hydrochloride and sodium methoxide in methanol to provide pyrimidinol **334**. Treatment of **335** with POCl_3_ produced chloride **335** which reacted with phenol derivatives **336a**-**b** using potassium phosphate (K_3_PO_4_) in DMA to deliver the desired ethers **337a**-**b**.

Some of the biological results for the most potent compounds are included in the patent [120]. Though the most potent compound was not disclosed, several derivatives showed an EC_50_ between 20–30 nM against hMRGX1 inositol monophosphate. The most promising compound, **337b,** had a very low CL_int_ of ca. <1.80 and 6.47 μg/mL/min in mouse and human, respectively. Compound **337b** exhibited a low clearance through IV in a mouse PK of around 7.1 ± 1.3 mL/min/kg with a volume distribution of 9.5 ± 3.1 L/kg and oral bioavailability (F) of 60 ± 2.6%. The low intrinsic clearance with high oral exposure of **337b** would allow for lower dose quantity/frequency while achieving therapeutic levels of target engagement. Compound **337b** also exhibited a very high total brain concentration of C_total brain_ = 64,100 ± 22,700 nM with the K_p,uu_ around 0.568 ± 0.165 for a 100 mg/kg single oral dose. The K_p,uu_ was indicative of good penetration into the CNS, suggesting that an active transport mechanism was not operative in mouse brain tissue.

## 7. Pyrimidine-Based Drugs for the Treatment of Diabetes Mellitus

A publication by Alam et al. evaluated pyrimidine as a core ring for agents to treat diabetes mellitus [122]. Diabetes mellitus is a metabolic disorder which is caused by hyperglycemia due to insufficient secretion of insulin, or resistance to insulin, or both. The number of people affected by diabetes will reach around 643 million by 2030 and 783 million by 2045, an increase of over 8% per year [123]. In this work, researchers tried to couple thiazolidinedione rings with a pyrimidine derivative for insulin resistance in peroxisome proliferator-activated receptor-γ (PPAR-γ) peripheral tissues. PPAR-γ tissue enhances insulin formation and displays prominent antihyperglycemic activity without causing hypoglycemia [124]. The team resorted to an in-silico modelling method for docking the core, and then designed, synthesized, and evaluated the biological activity of each compound.

Assembly of the pyrimidines of interest is outlined in Figure 43. The first step involved a one-pot Biginelli reaction using a series of *p*-substituted aryl aldehydes **338**, ethyl cyanoacetate and thiourea in the presence of K_2_CO_3_ in ethanol to form 1,6-dihydro-2-mercapto-6-oxopyrimidine-5-carbonitriles **339**. Intermediates **339** underwent S-alkylation with isopropyl bromide and NaOH in methanol to provide **340**. Chlorination of **340** with POCl_3_ produced chloropyrimidine **341** which underwent S_N_Ar etherification with phenol-substituted thiazolidinone derivatives **342** in the presence Cs_2_CO_3_ to form drug candidates **343**.

The synthesis yielded 13 derivatives using different benzaldehydes and all were evaluated for biological activity. Screening procedures identified two compounds, **344** and **345**, which demonstrated very good oral glucose tolerance test results in vivo using streptozotocin-induced diabetic rats for 28 days, and they both reduced blood glucose levels significantly. The compounds caused a significant (*p* < 0.0001) decrease in blood glucose levels compared with the standard drug Pioglitazone (see Appendix A). Compounds **344** and **345** decreased the blood glucose levels to 145.2 ± 1.35 and 146.6 ± 0.81, respectively, compared with Pioglitazone (150.2 ± 1.06). The compounds also showed a significant (*p* < 0.0001) decrease in triglycerides, total cholesterol and low-density lipoprotein cholesterol and an increase of high-density lipoprotein cholesterol. The biochemical estimations of hepatoxicity using alanine transaminase, aspartate transaminase, and alkaline phosphatase, along with urea, creatinine, blood urea nitrogen, total protein, and lactate dehydrogenase, indicated that the levels were restored to normal by **344** and **345** in treatment groups compared with a diabetic control group. Histopathological investigations revealed a normal architecture of the pancreas, liver, heart, and kidneys following administration of **344**. Finally, compounds **344** and **345** did not show any toxicity to mice or cause an increase in body mass.

## 8. Addendum

While this manuscript was being written, several reviews appeared with content overlapping the material in this paper. The first, by the Farghaly team [125], presented an excellent survey of the patent literature from 1980–2021 as it pertains to pyrimidines as antiviral compounds. A second review, by the Roh group [126] outlined recent pyrimidine derivatives developed as antitubercular agents. Finally, Saleem et al. [127], provided a summary on pyrimidine-based drugs as antibacterials. These contributions are more detailed and comprehensive treatments of three of the topics covered in the current review which is limited to compounds studied during the past 2–3 years of developmental work.

## 9. Conclusions

Molecular dynamics modelling and a growing body of knowledge have revealed numerous new structures that can interact with key enzymes important to the etiology of many debilitating conditions. With the plethora of precursors available, pyrimidine drugs predicted to bind with these enzymes should be readily accessible for screening. Many of the pyrimidines cited have shown IC_50_ values in the nM range, exhibited favorable ADME properties, and demonstrated compelling pharmacokinetic/pharmacodynamic readouts to become successful new drug candidates for various health conditions. The greater potency of these agents should translate to lower doses, causing fewer side effects relative to current pharmaceuticals. In addition to their high potency, pyrimidine-based drugs often avoid off-target toxicity, including hERG, ion-channels, CYP450 inhibition and induction, and cytotoxicity toward normal cell lines, thus establishing a safety window for their use. Indeed, many of the most potent prototype compounds showed low toxicity in test animals and this would hopefully extend to humans. Additionally, several of the candidate molecules discussed in this article presented evidence that pyrimidines hybridized with other ring systems were often competent at overcoming the increasing problems associated with drug resistance. Though pyrimidines were used as anti-infective and anticancer agents in the past, the recent discoveries with this ring have broken boundaries and extended its functional role against conditions which are difficult to treat, including neurological disorders such as Alzheimer’s disease, Parkinson’s disease and chronic pain.

The research summarized in this review should convince the reader of the high potential of targets built around the pyrimidine core ring structure. Overall, pyrimidine-based drugs and hybrid structures appear to be some of the most promising drug candidates among new medicinal agents on the horizon. While pyrimidines are already an important substructure within many current therapeutic agents, it is likely they will only gain increasing importance as new medications become necessary to maintain a healthy global society.

## Data Availability

No new data were created or analyzed in this review article. Data sharing is not applicable to this review.

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
