# Peer review of "Recent Advances in Pyrimidine-Based Drugs"

_pharmaceuticals, 2024, doi:10.3390/ph17010104_

Round 1

Reviewer 1 Report

Comments and Suggestions for Authors

I cannot recommend the publication of this work in Pharmaceutics. This manuscript aims to provide an overview of Pyrimidines are the latest field that has great influence on the discovery and treatment of anti-infective drugs, anti-cancer drugs, immunology, immunooncology, nervous system diseases, chronic pain and diabetes. However, the article has significant deficiencies in logical coherence, explanations of principles, analysis of field advancements, and lacks clear author perspectives. As a result, it falls short of effectively facilitating readers comprehensive understanding of the latest developments in this field. Addressing the following specific issues is crucial in enhancing the quality of the manuscript:

1. Some writing in the manuscript should be standardized, such as 9.0 µM/mL of 81 lines.

2. H3Ra variant did not explain the relationship with diseases in detail when it appeared for the first time, but it was repeated in later times, which is inappropriate.

3. Some synthesis processes in the manuscript are too lengthy, so it is suggested to modify them.

4. The structure of the manuscript is clear, but the content is quite lengthy and unorganized. It is suggested to use charts to express it.

Comments on the Quality of English Language

Extensive editing of English language required

Author Response

I cannot recommend the publication of this work in Pharmaceutics. This manuscript aims to provide an overview of Pyrimidines are the latest field that has great influence on the discovery and treatment of anti-infective drugs, anti-cancer drugs, immunology, immunooncology, nervous system diseases, chronic pain and diabetes. However, the article has significant deficiencies in logical coherence, explanations of principles, analysis of field advancements, and lacks clear author perspectives. As a result, it falls short of effectively facilitating readers’ comprehensive understanding of the latest developments in this field. Addressing the following specific issues is crucial in enhancing the quality of the manuscript:

General Response:

This reviewer claims that extensive English language revisions are required.  This is absurd.  I am a native speaker and have written hundreds of published papers.  There is nothing wrong with the English usage in this manuscript.  I am more than willing to adjust any factual information if I have misinterpreted something, but the English is not a problem.

The reviewer appears to be reviewing our manuscript for another journal... Pharmaceutics. We wrote this paper for Pharmaceuticals.

  1. Some writing in the manuscript should be standardized, such as “9.0 µM/mL” of 81 lines.

Response:  The units are those given by the original authors.  I don't believe it is possible to standardize all of them.

  1. “H3Ra variant” did not explain the relationship with diseases in detail when it appeared for the first time, but it was repeated in later times, which is inappropriate.

Response:  H37Ra (ATCC 25177) and H37Rv (ATCC 27294) are different strains of M. tuberculosis.  They are commercially available and could be handled easily in a laboratory.  We have included the ATCC numbers for each variant. The virulence factor for these strains is in line with drug-resistant strains.  That's why many of the researchers in this area used these strains.

  1. Some synthesis processes in the manuscript are too lengthy, so it is suggested to modify them.

Response:  General syntheses are included for the chemists who read the manuscript.  More detail is given, and reagent abbreviations included, for those not focused on the chemistry.  The descriptions of the syntheses are not long-winded and do not significantly add to the length of the paper.  We were attempting to bridge the fields of organic chemistry and pharmacy to make it an interesting read for researchers in both fields.

  1. The structure of the manuscript is clear, but the content is quite lengthy and unorganized. It is suggested to use charts to express it.

Response:  The manuscript is organized according to the disease being studied.  Charts would be appropriate if we were surveying all the known pyrimidine-based drugs, but we are focused on only the most recent.

Reviewer 2 Report

Comments and Suggestions for Authors

Nammalwar and Bunce presented a very interesting review that may be useful to both organic chemists and medicinal chemists. However, based on the contents of the manuscript, it is clear that the authors have placed the main emphasis on a review of antitumor agents (section 3 covers 61 references), while antibacterial (section 2.1 covers 11 references) and antiviral (section 2.3 covers 10 references) activities were considered rather sparingly. To compensate for this shortcoming, it would be necessary to add several literature references to recent reviews on antibacterial, antituberculosis (https://doi.org/10.1007/s11172-019-2686-x; https://doi.org/10.1016/j.ejmech.2022.114946; https://doi.org/10.1016/j.ejmech.2023.115701; etc.) and antiviral activities (https://doi.org/10.2174/1389557523666221220142911; https://doi.org/10.21608/aprh.2022.144745.1180; etc.).

The authors also need to carefully read the review again and correct typos in the reaction schemes (incorrect substituents and subscripts).

Finally, on page 2, the authors have written that the review covers the last two years, while based on their list of references, this period was estimated at the last two decades.

Author Response

Nammalwar and Bunce presented a very interesting review that may be useful to both organic chemists and medicinal chemists. However, based on the contents of the manuscript, it is clear that the authors have placed the main emphasis on a review of antitumor agents (section 3 covers 61 references), while antibacterial (section 2.1 covers 11 references) and antiviral (section 2.3 covers 10 references) activities were considered rather sparingly. To compensate for this shortcoming, it would be necessary to add several literature references to recent reviews on antibacterial, antituberculosis.

Response:  Over the past 2-3 years, research in this area has been dominated by studies on the treatment of cancer.  I believe this is where most of the research funding is directed.  Thus, the review leans heavily in this direction.

Response:  We have added an Addendum citing several of the references which appeared during the preparation of our manuscript.

https://doi.org/10.1016/j.ejmech.2022.114946: included in the addendum (pyrimidine based antituberculars);  this paper appeared in 2023.

https://doi.org/10.1016/j.ejmech.2023.115701: included in the addendum (pyrimidine-based antibacterials); this paper appeared in 2023.

https://doi.org/10.2174/1389557523666221220142911: included in the addendum 1980-2021 patent review (pyrimidine-based antivirals); this paper appeared in 2023.

https://doi.org/10.1007/s11172-019-2686-x: not included; describes earlier work; this paper appeared in 2019.

https://doi.org/10.21608/aprh.2022.144745.1180: not included as it only discusses syntheses of pyrimidines used as antivirals–nothing about activity.

The authors also need to carefully read the review again and correct typos in the reaction schemes (incorrect substituents and subscripts).

Response:  A general statement like this is not useful. The reviewer may have had trouble following some of the generalized synthetic schemes, but I do not believe there are a significant number of errors.

Finally, on page 2, the authors have written that the review covers the last two years, while based on their list of references, this period was estimated at the last two decades.

ResponseL  The references from the past have been included to provide background on certain targets and structures.  The current review primarily discusses work done only during the past 2-3 years.

Reviewer 3 Report

Comments and Suggestions for Authors

Dear Authors,

The theme you chose for your manuscript is interesting especially for chemists working in the medicinal chemistry area of interest. I congratulate you for the hard work involved in writing a review. 

I've read carefully your paper and I have some comments/suggestions:

- in lines 88-89, you write "compounds 20". I suggest using more numbers, due to the fact that are more compounds, not only one or using letters, for ex. 20 a-x. Also, in Scheme 1, it is not clear who are R1 and R2, therefore to which type of general structure correspond the compounds 23 and 24.

- what do you mean by "afforded targets 40" in line 201?

- also, what "exhibited 3-30-fold" in line 227 stands for?

An important aspect that is not developed in the paper is the mechanism of action of all compounds. I suggest insisting on this aspect and also on molecular docking. 

I suggest using bigger chemical structures.

The references are well-chosen.

Author Response

- in lines 88-89, you write "compounds 20". I suggest using more numbers, due to the fact that are more compounds, not only one or using letters, for ex. 20 a-x. Also, in Scheme 1, it is not clear who are R1 and R2, therefore to which type of general structure correspond the compounds 23 and 24.

Response:  R1 and R2 are shown in the generalized syntheses–this is common in organic chemistry papers.  However, the compounds in the box below (prepared by this general synthesis) are the most active and the focus of the discussion.  The word "compounds" used for a single number (as in Scheme 1) indicates all compounds with structures like 20 but with a different R group.  In some of the manuscripts, 50-75 compounds were prepared and tested.  We believe adding additional numbers and letters would be more confusing.

- what do you mean by "afforded targets 40" in line 201?

Response:  See above response.

- also, what does "exhibited 3-30-fold" in line 227 stand for?

Response:  This has been changed to 3- to 30-fold.

- An important aspect that is not developed in the paper is the mechanism of action of all compounds. I suggest insisting on this aspect and also on molecular docking.

Response:  The mechanism of action of these drugs in the treatment of these different diseases would require writing a book for each disease.  Our review surveys recently published manuscripts and does not attempt to design new compounds.  If the manuscripts included in our review had designed compounds with modelling studies, we could have included this in our review.  However, most of the papers did not include such studies so we have not highlighted this aspect in our article. 

- I suggest using bigger chemical structures.

Response:  The journal has space limitations and relatively large margins.  Any margin violations should be left to the editors.

Round 2

Reviewer 1 Report

Comments and Suggestions for Authors

This manuscript aims to provide an overview of long-acting injectable antipsychotics provide an effective treatment option for individuals with persistent mental illness.It is enough to publish through revision.

Comments on the Quality of English Language

Extensive editing of English language required